

# Anatomy of Arctic and Antarctic sea ice lows in an ocean–sea ice model

Benjamin Richaud[1], François Massonnet[1], Thierry Fichefet[1], Dániel Topál[1], Antoine Barthélemy[1], and David Docquier[2]

[1]Earth and Climate Research Center, Earth and Life Institute, Université catholique de Louvain, Louvain-la-Neuve, Belgium
[2]Dynamical Meteorology and Climatology Unit, Royal Meteorological Institute of Belgium, Brussels, Belgium

**Correspondence:** Benjamin Richaud (benjamin.richaud@uclouvain.be)

**Abstract.**

Sea ice has exhibited a number of record lows in both hemispheres over the past two decades. While the causes of individual sea ice lows have already been investigated extensively, no systematic comparison across events and hemispheres has been conducted in a consistent framework yet. Here, the global standalone ocean–sea ice model NEMO4.2-SI[3] at 1/4° resolution

is used to decompose the sea ice mass budget. We separate the relative contributions of sea ice melt versus growth and thermodynamic versus dynamic processes, both from a climatological perspective and for selected individual years. The seasonal cycles of Arctic and Antarctic ice mass fluxes show similarities, such as the prevalence of basal growth and melt in the mass budget. The long-term evolution of the mass budget terms reveals an increased importance of basal melt in both hemispheres, at the expense of surface and lateral melt. Regarding sea ice lows, the model indicates that the Arctic 2007 anomaly was chiefly

caused by dynamic factors, while the Arctic 2012 event was rather explained by thermodynamic factors. The Antarctic 2022 event was partly driven by a strong interplay between dynamic and thermodynamic processes. Regarding the Antarctic winter 2023 event, it was characterized by a notable lack of basal growth. This study emphasises the dominance of processes at the ice-ocean interface in driving the ice mass evolution at all time scales considered here, and highlights the potential of the ice mass budget decomposition to further our understanding of the evolution of polar regions in a changing climate.

## 1   Introduction

Polar regions have long been recognized as critical elements of the climate system. The Arctic and Antarctic regions are, however, not responding identically to the anthropogenic forcing. As an illustration, the Arctic minimum (September) sea ice cover (which is a principal indicator of the high-latitude boreal climatic state) has lost 44 % of its extent since the beginning of satellite observations (Docquier et al., 2024). Other ice parameters are also experiencing strong negative trends, with multiyear

ice older than 4 years having virtually disappeared since the 2010s (Kwok, 2018; Serreze and Meier, 2019) and annual sea ice thickness having experienced a 65 % reduction between 1975 and 2012 (Lindsay and Schweiger, 2015). The negative trends in sea ice extent, age and thickness are significant all year round, but are more evident in summer than in winter (Perovich and Richter-Menge, 2009), and the declines have been further strengthened in the past decade for most ice properties, except sea





ice extent (Meier and Stroeve, 2022). Superimposed on these long-term changes, interannual-to-decadal variability amplifies
the year-to-year fluctuations around the climatological conditions and modulates the slope of the trend (Swart et al., 2015;
Serreze and Stroeve, 2015). This variability has been increasing, in agreement with its expected evolution in a warmer climate
(Goosse et al., 2009), and has generated several record ice extent lows in recent years. The observed rapid Arctic sea ice loss
is explained by a combination of a strong response to the anthropogenic forcing, numerous positive feedbacks, and internal
variability (Goosse et al., 2018; Baxter et al., 2019). As a consequence, the background warming is regionally amplified, a
phenomenon known as the Arctic Amplification (Serreze and Francis, 2006), which leads to a regional surface temperature
increase at 3 to 4 times the rate of global warming (Rantanen et al., 2022). By contrast, Antarctic sea ice extent has seen no
change or even some slight increase from 1979 to 2015, with strong spatial heterogeneity (Turner et al., 2015; Hobbs et al.,
2016). Since 2016 however, the ice extent has started to decline (Eayrs et al., 2021; Purich and Doddridge, 2023). Ice volume
has followed similar variability as sea ice extent (Massonnet et al., 2013; Liao et al., 2022). The recent decrease since 2016 is
co-occurring with an observed increase in subsurface oceanic heat content and a potential shift in the mean state is currently
investigated (Zhang et al., 2022; Purich and Doddridge, 2023; Hobbs et al., 2024).

The interaction between long-term trends and interannual variability can result in the occurrence of sea ice lows, defined here
as instances when the sea ice extent becomes significantly lower than the linear trend line and exhibits a noticeable decrease
compared to the previous year's value. Several sea ice lows have been identified and described in the scientific literature. In the
Arctic, the sea ice extent experienced an unprecedented loss in boreal summer 2007. This previously unseen situation raised
concerns about the future of Arctic sea ice (Perovich and Richter-Menge, 2009). Most of the ice concentration anomaly was
located in the Pacific side and was attributed to preconditioning, anomalous winds, and ice-albedo feedback (e.g. Shimada
et al., 2006; Zhang et al., 2008). Low cloud coverage was initially suggested as a potential contributor but was quickly ruled
out (Schweiger et al., 2008), and the increased Pacific heat inflow has been suggested to be of minor influence, though it
could have triggered the onset of ice melt and a subsequent ice-albedo feedback (Perovich et al., 2008; Kauker et al., 2009;
Woodgate et al., 2010). Five years later, in summer 2012, the Arctic experienced another sea ice low, which remains the lowest
extent recorded so far (Francis and Wu, 2020). The bulk of the ice loss was located this time in the Atlantic side and on the
Siberian shelves. The main reasons for this ice low have been identified as preconditioning and higher than usual atmospheric
temperatures (Parkinson and Comiso, 2013; Guemas et al., 2013). A strong summer cyclone occurring in August was also
suggested to have played a role (Simmonds and Rudeva, 2012; Lukovich et al., 2021), though the impact of this cyclone on ice
loss has also been argued to be minimal (Zhang et al., 2013; Guemas et al., 2013).

On the other side of the planet, a few sea ice lows have also raised attention in recent years. In 2016, in early austral spring
(September), Antarctic sea ice melted faster than normal, resulting in the lowest minimum ice extent at that time, in austral
summer 2017. The ice loss anomaly was circumpolar, but was particularly noticeable in the Weddell and Ross Seas (Turner
et al., 2017), with a polynya observed over Maud Rise contributing to the negative anomaly in the Weddell Sea (Turner et al.,
2020). The origins of this sea ice low remain uncertain, but the event was subsequent to a strong ENSO event, a record negative
Southern Annular mode (Stuecker et al., 2017; Turner et al., 2017; Mezzina et al., 2024), and an ocean subsurface warm
anomaly (Zhang et al., 2022). This temperature anomaly has persisted until now (Purich and Doddridge, 2023). Mirroring what



happened in the Arctic, another sea ice low occurred five years later, in austral summer 2022 (Raphael and Handcock, 2022).

Most of the anomaly was located in the Ross Sea, with some contribution of the Weddell Sea. The ice loss was attributed to a record deepening of the Amundsen Sea Low, generating strong northward winds over the Ross Sea, creating a coastal polynya, and exporting sea ice offshore into warmer waters (Turner et al., 2022; Wang et al., 2022; Yadav et al., 2022). Yet, arguably the most intriguing evolution of the Southern Ocean sea ice cover happened in austral winter 2023, when the sea ice extent was 2.4 million squared kilometres (five standard deviations) lower than the climatological average (Ionita, 2024; Espinosa et al.,

2024). Negative anomalies were observed around most of the Antarctic continent and were particularly marked in the Weddell and Ross Seas and in the Indian sector (Gilbert and Holmes, 2024). This anomalously slow winter expansion of the sea ice cover followed a record low sea ice extent in austral summer 2023 and preceded an austral summer and winter 2024 of anomalously low sea ice extent as well. Causes are still under investigation, but record high sea surface and subsurface temperatures have been observed in regions with reduced sea ice cover (Espinosa et al., 2024). A zonal wave number 3 atmospheric pattern could

also have favoured heat and moisture advection over areas with most ice loss (Ionita, 2024). Warm conditions in the Southern Ocean that developed prior to 2023 can explain over two thirds of the observed 2023 anomaly (Espinosa et al., 2024). While some causes such as preconditioning seem to be a common thread through all those events, the diversity of candidate drivers is striking, though expected: anticyclonic conditions in 2007 or summer cyclone in 2012, negative Southern Annular Mode in 2017 or enhanced Amundsen Sea Low in 2022, anomalous oceanic heat inflow or atmospheric temperatures, subsurface

conditions, anomalously thin sea ice state, local ice-albedo feedback, or anomalous wind forcing, etc. One hypothesis of the present work is that sea ice lows are not triggered by one single mechanism, but instead result from the combination of several drivers being anomalous at the same time.

Due to the strong non-linearities of the climate system, an extreme low in sea ice extent can have disproportionate impacts. The ice-albedo feedback, for example, would lead to more solar absorption during a sea ice low, delaying the following freeze-

up season and leading to thinner ice, as documented for the 2007 event (Timmermans, 2015). Such events could therefore have a lasting impact on the sea ice state. Moreover, sudden retreats of sea ice have been shown to drive an increase in climate extremes in regions surrounding the Arctic Ocean (Delhaye et al., 2022). A proper understanding of the mechanisms leading to extreme sea ice events is therefore critical to anticipate impacts at larger temporal and spatial scales. Yet, all the aforementioned events have been investigated with a range of data sources, including in-situ and remote observations as well

as numerical models and reanalysis data (e.g. Kauker et al., 2009; Woodgate et al., 2010; Yadav et al., 2022). The large diversity of identified mechanisms listed previously could therefore also be a consequence of those different approaches and data sources. To the authors knowledge, no known study has provided to date a consistent framework to devise similarities and differences between years, seasons, and hemispheres, though some studies have compared a couple of events together, mostly from the same hemisphere (e.g. Mezzina et al., 2024). In order to disentangle the causes leading to the sea ice lows, a

coherent comparison of the sea ice lows is necessary. Ice areal coverage is now routinely monitored year round from space. However, it conveys only part of the more fundamental sea ice state that is better captured by sea ice mass or sea ice volume. The mass and volume provide more information than the areal coverage for the heat budget and fluxes in polar regions, and for atmosphere-ice-ocean interactions. Moreover, while sea ice thickness follows a similar trend to sea ice extent, it does not



necessarily follow its interannual variability or seasonality (e.g. Kwok, 2018; Landy et al., 2022). Investigating the ice mass
budget can therefore provide better insight on the state of polar climates than focusing on ice concentration.

To this aim, we offer a decomposition of the sea ice mass budget as provided by an ocean-sea ice numerical model. We
investigate the mean state and long-term trends of the sea ice mass fluxes in both hemispheres, and turn to mass flux anomalies
to compare four different sea ice lows, namely the Arctic 2007 and 2012 boreal summers, and the Antarctic 2021-2022 austral
summer and 2023 austral winter. This approach allows to disentangle the respective roles of dynamic and thermodynamic terms,
including their spatial and temporal variabilities. It also highlights the dominant role of the ice-ocean interface in driving sea
ice lows, along with preconditioning. Further analysis provides more insight on the oceanic role through increased oceanic heat
content. Section 2 introduces the ocean-sea ice model and the sea ice mass balance relied upon in this study. Section 3 describes
the climatological seasonal cycle and the trends of the different mass fluxes and compares both hemispheres, highlighting the
good agreement of the model with the current scientific literature. Section 4 focuses on the four above-mentioned case studies,
comparing years, hemispheres, and seasons to paint a general picture of dominating drivers. Section 5 summarises the results
and provides a perspective on similarities and differences between all those events, and on the potential for future sea ice lows.

## 2 Methods

### 2.1 Model description and evaluation

In this study, we use the Nucleus for European Modelling of the Ocean (NEMO) version 4.2.2 (Madec et al., 2023), which
includes the Sea Ice modelling Integrated Initiative (SI$^3$) model (Vancoppenolle et al., 2023). The model is configured on
the global eORCA tripolar grid, using a 1/4° nominal horizontal resolution, which makes it eddy-permitting. The vertical
grid is set up on 75 z-coordinate levels, with 8 levels in the first 10 m (24 in the first 100 m), allowing to properly resolve
the shallow summer mixed layer in the Arctic Ocean. NEMO is a primitive equation model using a three-dimensional, free-
surface, hydrostatic, Boussinesq-approximation approach. SI$^3$ is a dynamic-thermodynamic continuum sea ice model, using
a two-dimensional elastic-viscous plastic rheology for dynamics and a one-dimensional energy- and salt-conserving method
for thermodynamics. In the configuration used here, a subgrid-scale distribution of ice thickness is discretised into five cat-
egories, and each category is further divided into two ice layers and two snow layers. Sea ice dynamic processes simulated
by SI$^3$ include horizontal advection, rheology, and ridging/rafting. Regarding the thermodynamic processes, the model in-
cludes energy conservation, an explicit representation of sea ice salinity, solar radiation penetration and transmission, a surface
albedo function of ice temperature, thickness, snow depth, cloud fraction, and melt ponds, lateral melting through ice floe size
parametrisation, snow-to-ice conversion, and level-ice melt ponds including frozen lids.

The ocean and sea ice model are driven by atmospheric fields extracted from the ECMWF ReAnalysis v5 (ERA5, Hersbach
et al., 2020). The ocean model starts from rest from 3D temperature and salinity fields from the World Ocean Atlas 2018
(WOA18 Garcia et al., 2019; Locarnini et al., 2018; Zweng et al., 2019), is run over the period 1960 to 2023, and its results
are analysed over 1979 to 2023. A sea surface salinity restoring towards the WOA18 data is implemented in the form of a
corrective surface freshwater flux, with a strength corresponding to a time scale of 227 days for a mixed layer depth of 50 m.




Such a restoring is commonly applied in forced ocean models, which do not have a closed freshwater budget, in order to prevent unrealistic drifts of the sea surface salinity. This flux is scaled by a factor of $1 - c$, with $c$ denoting the sea ice concentration, to disable the restoring in ice-covered areas and avoid interfering with the impacts of sea ice formation or melting on salinity. Ice
variables used in this study, including ice concentration, thickness, volume, and all mass fluxes described below, are written out as daily outputs, while ocean-related properties such as salinity and temperature are written with a monthly frequency.

In order to evaluate the model, the results are compared to the satellite-based sea ice extent calculated from the NOAA/N-SIDC Climate Data Record (CDR) of Passive Microwave Sea Ice Concentration, version 4 (Meier et al., 2021). The sea ice extent is calculated as the total area of grid cells with ice concentration above 15 %. The climatological seasonal cycle, calcu-
lated over the 1979-2023 period (see below for climatology calculation description) shows a clear negative bias for the model relative to the CDR observations, at least in the Arctic (Figure 1.a and c). This bias is visible in summer for both hemispheres, with a 2 millions km$^2$ underestimation in the Arctic and a 1.3 millions km$^2$ underestimation in the Antarctic. The modelled winter sea ice extent in the Antarctic is higher than in the observations, leading to an annual mean close to the observation-based mean. While these biases are a real concern (and are likely related to a warm bias in the atmospheric forcing as reported
by Zampieri et al., 2023), they are a second-order issue for the present study since we focus our analyses on anomalies rather than on the mean state. We discuss these aspects further in Section 5. In both hemispheres, the modelled winter extent peaks a few weeks later than in observations, but the timing of the summer extent minimum is consistent between both data sources. This leads to an overall shorter, more intense melting phase and longer growth season, and to an overestimation of the modelled amplitude of the seasonal cycle. The negative bias in summer is a known issue of the model (Rousset et al., 2015) and is likely
related to a positive air temperature bias in the ERA5 forcing set, due to lack of representation of snow over ice in the model used to produce ERA5 (Batrak and Müller, 2019). To investigate the interannual variability of the ice extent, the climatological seasonal cycle is removed from the data to obtain sea ice extent anomalies, and we focus on the month of minimum sea ice extent (Figure 1.b and d). For the remaining of this study, the month of minimum sea ice extent refers to September for the Arctic and February for the Antarctic. The sea ice extent variability is overall well captured by the model, at most time
scales. The correlation between observations and model minimum extent anomalies is higher for the Arctic ($r = 0.93$) than for the Antarctic ($r = 0.51$) when considering the entire time series, but increases in the Antarctic when considering the last two decades ($r = 0.76$). Decadal oscillations in the Arctic are visible and in phase in both model and observations, and the Antarctic high extent from 2012 to 2015 followed by the sharp drop to lower values until present is also well simulated by the model. But the amplitude of the variability is overestimated, especially in the Antarctic. Despite that, many of the sea ice lows
of interest are reasonably well captured, including the 2012 low in the Arctic and the 2022-2023 lows in the Antarctic. The events of 2007 in the Arctic and 2017 in the Antarctic are also simulated, but are followed in the model by a year of lower ice extent, making them less striking in the model than in observations. A few other lows are visible, such as the Arctic 1990 or 2003, which are also visible in observations but with a smaller magnitude, or the Antarctic 1999 which is not matched by any ice loss in the observations. Overall, the realistic simulation of interannual variability lends confidence in the use of this
tool for the analysis of individual sea ice lows and we expect the model to provide reliable information about the sources and





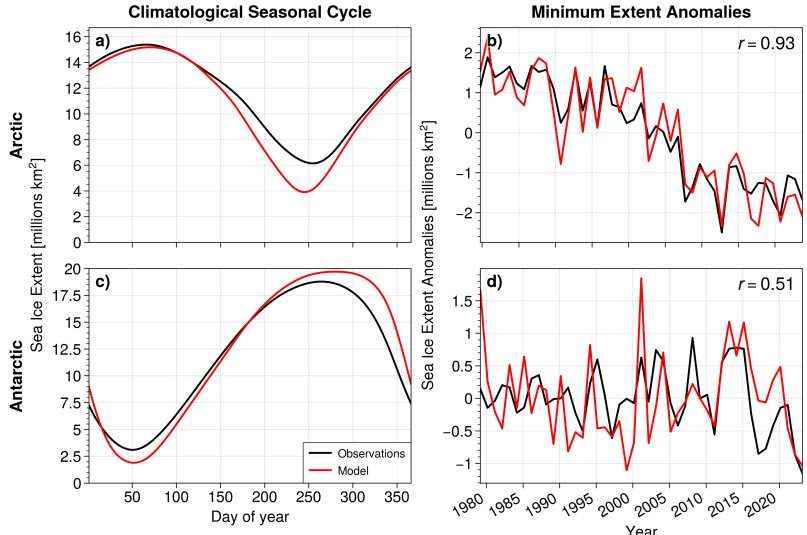

**Figure 1.** Comparison of satellite-based observational product NOAA/NSIDC Climate Data Record (CDR, black) versus model (red) sea ice extent. Climatological seasonal cycle of sea ice extent for (a) Arctic and (c) Antarctic. Minimum sea ice extent anomalies relative to the climatological seasonal cycle for (b) September in Arctic and (d) February in Antarctic; correlations between observations and model are given in the top-right corner of each panel.

sinks of sea ice mass during these events. A comparison to previously published studies is provided when analysing specific ice properties and sea ice lows.

## 2.2 Sea ice mass balance

Although sea ice extent is a convenient and widely monitored diagnostic of sea ice change, it provides only a limited view of
the overall sea ice state. In contrast, sea ice mass offers a more comprehensive and physically meaningful understanding of the sea ice changes, as it accounts for both areal and thickness variability at each grid point. However, it cannot be observed at similar spatial and temporal scales. The model is therefore a necessary step to further our understanding of sea ice lows. In the model the instantaneous change in sea ice mass (i.e., the ice mass tendency) is the result of the sum of sources and sinks of thermodynamic and dynamic origins (Figure 2):

$$\frac{dM_i}{dt} = F_i^{\text{growth}} + F_i^{\text{melt}} + F_i^{\text{dynamics}} \tag{1}$$

with $M_i$ referring to the sea ice mass over the region of interest (in kg or kg m$^{-2}$ of grid cell area when expressed at a given grid point), $F_i^{\text{growth}}$ indicating the mass flux due to thermodynamic processes leading to ice mass gain, $F_i^{\text{melt}}$ designating the mass flux due to thermodynamic processes leading to ice mass loss, and $F_i^{\text{dynamics}}$ referring to the transport term of sea ice mass in or out of the region of interest (all fluxes in kg d$^{-1}$). Integrated over a hemisphere, the transport term is necessarily zero, as
what is exported from a grid cell is imported into a neighbouring grid cell. The thermodynamic component associated with ice



gain can be further decomposed into specific processes:

$$F_i^{\text{growth}} = F_i^{\text{frazil}} + F_i^{\text{basal g.}} + F_i^{\text{snow-ice}} + F_i^{\text{ridging}} \tag{2}$$

with $F_i^{\text{frazil}}$ referring to the mass flux due to initial formation of frazil ice in open water, $F_i^{\text{basal g.}}$ indicating the flux due to basal growth once the ice reaches a thickness of 0.1 m, $F_i^{\text{snow-ice}}$ designating the flux due to conversion from snow to ice when the ice-snow interface is depressed below sea level, and $F_i^{\text{ridging}}$ referring to the ice formation due to freezing of seawater trapped within porous ridges. It is worth noting that snow-to-ice conversion is a process that typically occurs when seawater floods the surface of sea ice and freezes, either due to increased snow load or basal melt depressing the snow-ice interface below the sea surface. For this reason, basal melt and snow-to-ice are correlated, as will become clearer in Section 4. The other thermodynamic component collects all the mass fluxes associated with sea ice melt:

$$F_i^{\text{melt}} = F_i^{\text{basal m.}} + F_i^{\text{lateral}} + F_i^{\text{surface}} + F_i^{\text{pond}} + F_i^{\text{sublimation}} \tag{3}$$

with $F_i^{\text{basal m.}}$ showing the mass flux due to basal melt at the ice-ocean interface, $F_i^{\text{lateral}}$ referring to the mass flux due to melt at the lateral interface to reduce ice concentration, $F_i^{\text{surface}}$ describing the flux due to surface melt at the ice-atmosphere interface, $F_i^{\text{pond}}$ indicating the flux due to melt pond lid melting, and $F_i^{\text{sublimation}}$ denoting the flux due to sublimation of ice at the surface. Those last two terms and the porous ridge growth term are negligible (see Table 1 and Figure 3) and will not be shown nor discussed for the rest of the study. Note that they are nonetheless accounted for in the calculations of the total mass fluxes, to ensure that the budget is closed. This ice mass balance decomposition has been conducted in other studies for the Arctic region (e.g. Keen et al., 2021) or the Antarctic (e.g. Li et al., 2021) from a climatological or long-term evolution perspective, and we attempt to follow similar conventions when possible. To that effect, fluxes are always considered from the point of view of the sea ice itself, meaning a positive flux corresponds to a mass gain (due to growth or import), while a negative flux indicates a mass loss (due to melt or export).

Anomalies of each term are estimated by removing the climatological seasonal cycle. The latter is calculated using a double smoothing over the 1979-2008 period, following the WMO recommendation of a 30 year long period (WMO, 2017) and starting with the first complete year of satellite coverage. A 11-day window centred around the day of year of interest is used to calculate the mean, then a 31-day moving average is used to smooth the signal a second time. Trends are also calculated for each term, to evaluate the evolution of mass fluxes over time. Those trends are computed using a linear fit over the period of interest (1979-2023). The seasonal cycles, trends, and anomalies are calculated for the whole Arctic and Antarctic region, but are also further decomposed into sectors to identify spatial heterogeneity. For the Arctic, 7 sectors are used, based on the definitions from Koenigk et al. (2016). For the Antarctic, sectors based on ice variability rather than geographical considerations are used, as provided by Raphael and Hobbs (2014).[1]

---

[1]Code and sector definitions available at https://forge.uclouvain.be/BenjaminRichaud/polarsectors



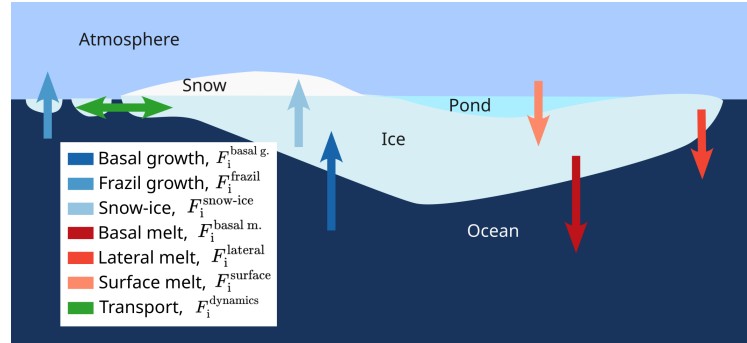

**Figure 2.** Schematic of most important ice mass fluxes considered in the NEMO-SI$^3$ ocean-sea ice model. An upward arrow indicates an (always positive) mass flux due to growth, while a downward arrow indicates an (always negative) mass flux due to melt. The horizontal arrow indicates dynamics-induced mass flux which can either be positive or negative.

## 3 Mass fluxes: climatology and trends

### 3.1 Climatology

The seasonal cycle of ice mass fluxes highlights the dominating terms driving the growth and melt of sea ice (Figure 3). In the Arctic, basal growth is by far the dominant flux leading to ice formation, explaining close to 75 % of the ice mass gain (Table 1, numbers in parenthesis). This is also the case when looking at regional patterns, except in the Greenland Sea sector, where basal growth accounts for less than half of the total growth, compensated by snow-to-ice conversion, which amounts to a third of total ice growth (Figure 3). Apart for that latter sector, frazil ice formation is the second most important flux and can explain 10 to 20 % of ice mass gain, especially in sectors of seasonal ice cover such as the Chukchi-Bering Seas and Barents-Kara Seas sectors. Overall, for the Arctic, mass gains related to snow-to-ice and porous ridging are negligible, in agreement with observations for the snow ice formation (Gani et al., 2019). Regarding ice melt, the dominating term is basal melt, explaining 62 % of mass loss, followed by surface melt with 25 %. Both fluxes occur in the summer months, except basal melt that can be significant in the Labrador Sea-Baffin Bay, Greenland Sea and Barents-Kara Seas sectors in winter and spring, due to year-round transport of sea ice towards warmer waters in those areas, or inflow of warmer waters towards the ice edge in the case of the Barents and Greenland Seas. Surface melt tends to start slightly later than basal melt, but peaks slightly earlier in most sectors. Lateral melt is much smaller than the other fluxes, except in the Greenland Sea sector where it can account for a third of the melt, once again due to the transport of sea ice towards warmer waters. Ice transport follows some known features, such as export of ice along the Transpolar Drift from the Chukchi and Siberian Seas into the Central Arctic and then into the Greenland Sea through Fram Strait (Figure 3.a, green lines). Pond melt and sublimation are always negligible. Those results match very well in terms of relative magnitudes and timing with those found in a Coupled Model Intercomparison Project phase 6 (CMIP6) model intercomparison (Keen et al., 2021) or an ice thickness distribution sensitivity experiment based on an earlier version of the same model as used here (Massonnet et al., 2019). The absolute magnitudes are slightly higher than those





reported in Keen et al. (2021). This could be due to a difference in the period of interest and to the overestimated amplitude of the seasonal cycle in ice extent in the model used here (Figure 1).

The overall picture in the Antarctic is similar, with the prevalence of basal growth and basal melt in the ice mass fluxes (Figure 3.b), accounting respectively for 48 % of ice mass gain and 87 % of loss (Table 1). Yet, some differences exist. For the ice formation, frazil ice formation is here again much smaller than basal growth, but snow-to-ice conversion is an important term in all sectors, representing nearly as much mass gain as basal growth (31 %) and being even the dominating term in the East Antarctic sector. It is worth noting that the timing of snow-to-ice conversion is different from the other ice growth terms, as it occurs during austral spring and summer. This is due to the fact that snow-to-ice conversion is triggered by surface flooding, which is more likely when basal melt lowers the freeboard. Surface melt is negligible in all sectors, in contrast to the Arctic: the thicker snow layer increases the albedo, reducing the impact of solar radiation on the surface heat input, and acts as a better insulator, reducing the thermal imbalance at the ice-snow interface that could result in surface melt. Lateral melt plays a similar role as in the Arctic, accounting for slightly more than 10 % of the total ice loss. The circumpolar current which should lead to an eastward ice transport is not clearly visible, as what comes in goes out in the sectors as they are defined (Figure 3.b). Nonetheless, transport is more important in the break-up season (spring and summer) than in other seasons, as one might expect from more mobile sea ice, with the exception of some ice exchange visible in winter between the Bellingshausen Sea and Weddell Sea sectors, when ice is forced through the Drake Passage. During the break-up season, some export of ice from the Weddell Sea sector to the King Håkon VII sector is visible, as well as from the Ross-Amundsen Seas sector into the adjacent Bellingshausen Sea and East Antarctic sectors. Moreover, some transport occurs northward, though not visible in the aggregated term. This explains the occurrence of basal melt year-round in all sectors, as ice is transported towards warmer regions in winter. The seasonality, and relative and absolute magnitudes of those terms are very close to those reported in another CMIP6 model intercomparison (Li et al., 2021). They are also in good qualitative agreement with the ice thickness distribution sensitivity study of Massonnet et al. (2019), though some quantitative discrepancies exist.

While the relative contributions of the different dominating terms show some spatial heterogeneity in the Arctic Ocean, they are more homogeneous in the Southern Ocean, with the overall seasonal cycle being representative of individual sectors. It is also interesting to note the similarities between the Greenland Sea sector and the Antarctic, with year-round basal melt due to equatorward export and reduced surface melt and increased snow-to-ice conversion due to higher rates of snowfall.

The climatological averages presented in this section mask the reality that the relative contributions of mass fluxes has likely changed over time, as suggested by the observed trends in sea ice extent. In the next section, we therefore take a broader perspective and inspect how these mass fluxes have evolved over time.

## 3.2 Trends

In order to calculate trends, we first integrate all growing and melting terms over a year (from September to August in the Arctic and from February to January in Antarctic, to match the sea ice thickness minimum), then normalize each term by the total annual ice growth or melt. This yields a percentage of ice growth due to each of the basal growth, frazil growth, or snow-to-ice conversion, and a percentage of ice melt due to basal, lateral, or surface melt, for each year. A linear trend for each



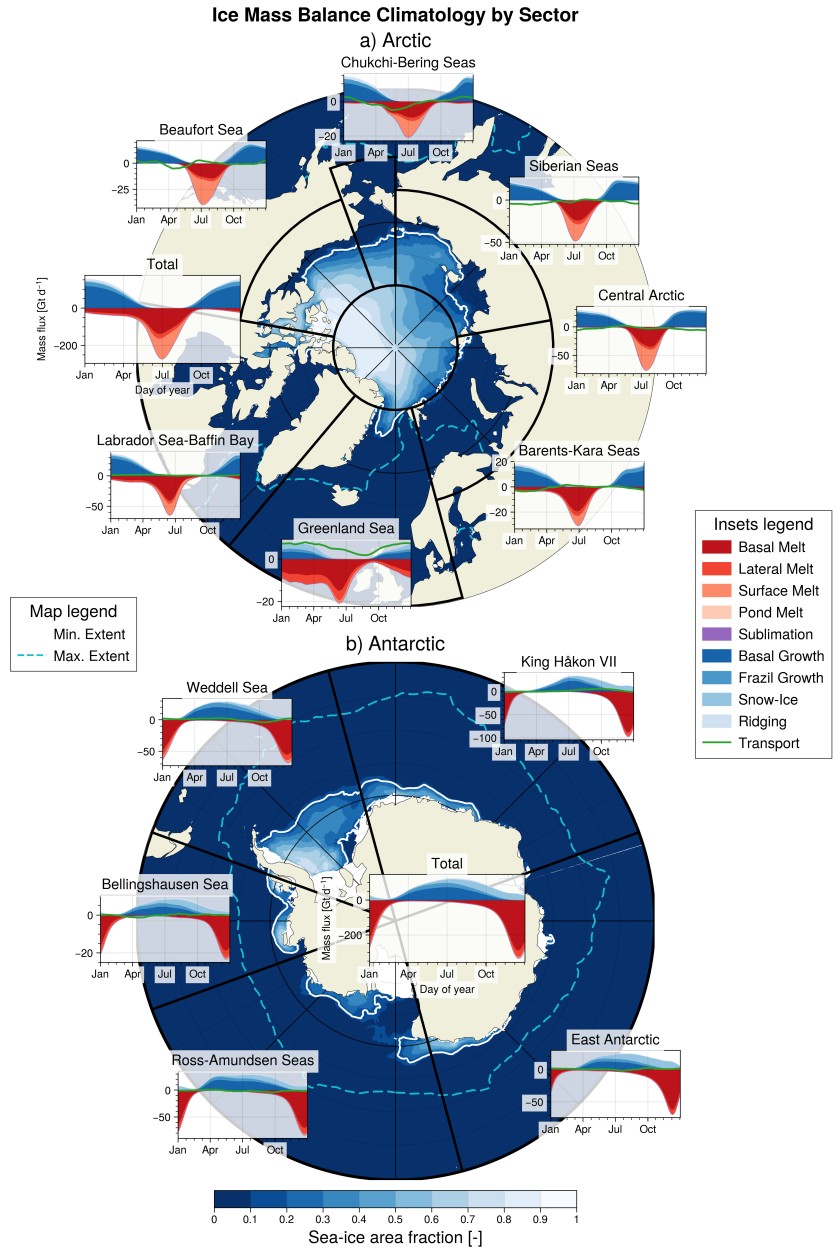

**Figure 3.** Climatological seasonal cycle of ice mass fluxes for total hemisphere and regional sectors, for Arctic (a) and Antarctic (b). Background maps show the climatological (1979-2008) average sea ice concentration during month of minimum extent (colours), minimum sea ice edge (white solid line) and maximum sea ice edge (cyan dashed line). Each panel provides the climatological seasonal cycle of ice mass fluxes as a function of the day of year, with each coloured area corresponding to a flux (see legend for details), for each sector delimited by the black boundaries on the map; the scales of the y-axes differ among panels. The ice mass transport is shown by a green line, as it can be both positive (ice import) and negative (ice export). The Total panels (left for Arctic, centre for Antarctic) show the climatological seasonal cycle for the whole hemisphere.





term can then be calculated over the period of interest (1979-2023), along with a *p*-value to test significance (Wald Test with t-distribution).

In both hemispheres and in most sectors, basal melt has been increasing to the detriment of surface and lateral melts (Table 1). In the Arctic, this is particularly true when discarding the Greenland Sea and Labrador Sea-Baffin Bay sectors, and is consistent with observations that documented the shift from surface to basal-driven melt in the Beaufort Gyre and around the North Pole
(Carmack et al., 2015). The lack of in-situ observations in the Southern Ocean prevents a similar comparison, but the increased basal melt at the expense of surface and lateral melts is the strongest in the Bellingshausen Sea sector, which has experienced a negative trend in sea ice extent over the satellite period (e.g. Cavalieri and Parkinson, 2008; Maksym, 2019). A longer ice-free season would increase ocean heat uptake and consequently basal melt. The ocean heat content has also been increasing, especially in the Arctic, as well as the oceanic heat transport at Arctic gateways, and could be a dominating driver for these
trends (Timmermans, 2015; Docquier et al., 2021).

Some ice growth terms also exhibit trends, with basal growth increasing to the detriment of ridging in the Arctic but with strong disparities: at sectorial level, the only significant trend for basal growth is in the Barents-Kara Seas sector, and is actually negative, contrarily to the hemispheric trend. This negative trend is due to a change from perennial to seasonal ice cover and is well reflected in the increase in frazil growth, related to a wider open water area to refreeze at the end of the melting season. The
longer open-water season also leads to further atmospheric heat being stored in the upper ocean, delaying the freeze-up season and reducing subsequent basal growth (e.g. Timmermans, 2015). For the same reasons, frazil growth has been increasing in the Labrador Sea-Baffin Bay sector. The lack of trend for frazil growth at the Arctic level is surprising at first, as the seasonal ice cover has been expanding due to a faster decrease in summer ice extent compared to winter ice extent. A stronger decrease in absolute basal growth explains this lack of frazil growth relative trend. In the Antarctic, basal growth shows a relative decline
at an hemispheric level and in the Weddell Sea and King Håkon VII sectors, compensated by a relative increase in snow-to-ice conversion. The relation is opposite in the East Antarctic sector, where an increase in basal growth is co-occurring with a decrease in snow-to-ice conversion. Yet, the causality is likely not between those terms, but rather between basal melt and snow-to-ice conversion, as mentioned earlier. The opposite sign of basal growth and melt trends could be related to increased snow precipitation better insulating the ice and to a warmer subsurface ocean increasing heat transfer in the growing season,
as both are seen in model forcing and outputs (not shown for snow precipitation, see Figure 6 for ocean heat content increase in the Southern Ocean).

Overall, similar trends are found in both hemispheres, with basal melt taking over surface and lateral melt, and spatially variable relative trends of basal growth within each hemisphere. The lack of clear trends before 2020 in the anomalies provided by Keen et al. (2021) prevents any easy comparison, but the absolute trends in the model used in the present study are of the
order of 1000 Gt over the run period, with a sign matching the long-term evolution of the CMIP6 models (Keen et al., 2021, their Fig. 12). Our model is therefore in agreement with the CMIP6 models and with in-situ observations for the Arctic, as mentioned earlier. No published estimate on the long-term evolution of the ice mass fluxes could be found for the Antarctic.





**Table 1.** Evolution of relative contributions of important mass budget terms to annual growth or melt, by sector and hemisphere. Linear trends are given in percentage per decade, and the number in parenthesis highlights the average yearly contribution of the term, in %. Residuals encompass porous ridging for growth, and sublimation and pond lid melt for melt. Bold font indicates trends that are considered as significant ($p$-value $< 0.05$).

| | Basal growth | Frazil growth | Snow-Ice conversion | Residuals (growth) | Basal melt | Lateral melt | Surface melt | Residuals (melt) |
|---|---|---|---|---|---|---|---|---|
| Barents-Kara Seas | **-1.2 (70)** | **1.0 (21)** | **0.6 (4)** | **-0.4 (5)** | **1.4 (72)** | -0.2 (15) | **-1.2 (12)** | **0.0 (1)** |
| Beaufort Sea | 0.3 (81) | 0.1 (12) | 0.0 (0) | **-0.4 (7)** | **1.9 (47)** | 0.1 (9) | **-1.9 (44)** | -0.0 (0) |
| Central Arctic | 0.4 (80) | **-0.3 (12)** | **0.2 (1)** | **-0.4 (7)** | 0.7 (55) | -0.1 (9) | -0.6 (36) | -0.0 (0) |
| Chukchi-Bering Seas | 0.3 (72) | -0.1 (19) | 0.1 (1) | **-0.3 (8)** | **1.4 (61)** | -0.3 (11) | **-1.1 (28)** | 0.0 (0) |
| Greenland Sea | 0.6 (47) | -0.3 (17) | -0.1 (34) | **-0.3 (2)** | 0.1 (68) | -0.0 (28) | -0.0 (4) | 0.0 (1) |
| Labrador Sea-Baffin Bay | -0.2 (76) | **0.3 (16)** | -0.0 (3) | **-0.1 (4)** | **0.7 (66)** | **-0.4 (12)** | -0.3 (21) | **0.0 (0)** |
| Siberian Seas | 0.4 (80) | -0.0 (14) | 0.0 (0) | **-0.4 (6)** | **1.3 (58)** | -0.1 (11) | **-1.3 (31)** | **0.0 (0)** |
| Total | **0.3 (75)** | -0.0 (16) | -0.0 (3) | **-0.3 (6)** | **0.6 (62)** | **-0.3 (13)** | -0.3 (25) | 0.0 (0) |
| Bellingshausen Sea | -0.1 (38) | **-1.1 (20)** | 1.0 (38) | 0.1 (5) | **1.1 (81)** | **-0.5 (16)** | **-0.5 (2)** | **-0.0 (1)** |
| East Antarctic | **0.6 (38)** | 0.1 (17) | **-0.8 (43)** | -0.0 (2) | 0.1 (88) | -0.1 (11) | 0.1 (1) | -0.0 (1) |
| King Håkon VII | **-1.1 (47)** | **-0.3 (16)** | **1.3 (35)** | **0.1 (2)** | **0.3 (90)** | -0.1 (8) | **-0.2 (2)** | -0.0 (1) |
| Ross-Amundsen Seas | -0.2 (52) | -0.1 (16) | 0.4 (30) | -0.0 (3) | 0.3 (85) | **-0.4 (13)** | 0.2 (2) | **-0.0 (1)** |
| Weddell Sea | **-1.9 (57)** | **0.8 (23)** | **1.1 (17)** | -0.0 (3) | **0.6 (83)** | 0.1 (12) | **-0.7 (4)** | -0.0 (1) |
| Total | **-0.6 (48)** | -0.0 (18) | **0.6 (31)** | 0.0 (3) | **0.4 (86)** | **-0.2 (11)** | -0.2 (2) | **-0.0 (1)** |

## 4 Case studies

The ice mass fluxes can then be used to dive into specific cases, namely the sea ice lows that occurred in both hemispheres. The consistent framework provided by the model opens the possibility of intercomparing events with each other. We compare two different sea ice lows in the same hemisphere (boreal summers 2007 and 2012 in the Arctic), two different sea ice lows in two different hemispheres (boreal summer 2012 in the Arctic and austral summer 2022 in the Antarctic) and two different sea ice lows in different seasons in the same hemisphere (austral summer 2022 and winter 2023 in the Antarctic).

### 4.1 Comparing two different years: Arctic 2007 and 2012

The 2007 Arctic sea ice low was characterized by large reductions in sea ice concentration and thickness on the Pacific side of the Arctic Ocean according to satellite observations (Perovich et al., 2008; Kauker et al., 2009). This feature is simulated by the model, with the strongest ice thickness anomalies in the Chukchi-Bering Seas and Siberian Seas (Figure 4.a). Some negative thickness anomalies are also visible along the southern fringes of the Beaufort Gyre. Positive thickness anomalies are visible along the ice edge towards the Central Arctic, suggestive of ridging and convergence in those areas, as expected in those more dynamical regions. However, the mass budget analysis offers another perspective on the causes of this 2007 anomaly. In the

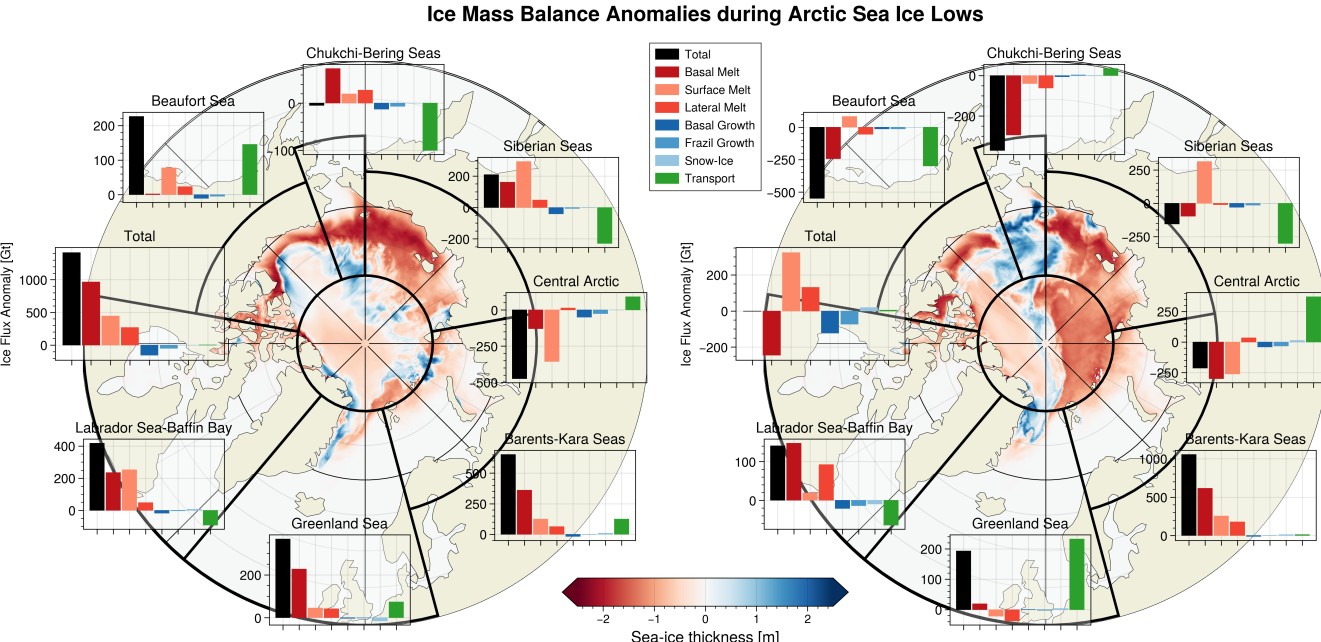

**Figure 4.** Arctic (a) 2007 and (b) 2012 sea ice lows. Spatial distribution for August ice thickness anomalies with respect to 1979-2008 climatology (background map) and ice mass budget anomalies integrated from May to August for each sector (as delimited by black lines) and for the total Arctic (inset panels; the scales of the y-axes differ among panels). A positive anomaly means an anomalous gain of ice mass, i.e. less ice melt or more ice growth than climatological conditions.

conventions used here, a positive anomaly means enhanced growth or reduced melt compared to climatology. At the Arctic scale, the net (total) mass flux integrated from May to August exhibits a strong positive anomaly, meaning that there was less melt in 2007 than during a typical year (because the growth terms are near zero during that period, Fig. 3). This may seem counter-intuitive at first, but the decomposition in sectors helps to better disentangle the underlying reasons. The bulk of this
melt deficit is due to a lack of basal and surface melt in the subpolar seas, namely the Labrador Sea-Baffin Bay, Greenland Sea, and Barents-Kara Seas sectors. This is due to preconditioning, as the sea ice mass anomaly in March 2007 was already strongly negative in those three sectors, implying less ice to melt than during a climatological year (supplementary materials, Figure S1). This induces a positive anomaly in the budget calculations. The Beaufort Sea sector also experiences a positive anomaly. This one is not due to preconditioning, but rather to less surface melt than usual and to increased ice import into the
sector, from the Chukchi Sea. The positive anomaly in the Siberian Seas sector is a different case, where export of ice out of the sector induces less surface and basal melt due to the reduced presence of sea ice; this is also the case in the Chukchi-Bering Seas sector, though the net outcome is zero. The only Arctic sector that exhibits a negative anomaly is the Central Arctic one, where enhanced surface and basal melts lead to reduced volume of sea ice. When excluding the Atlantic-side subpolar seas (Labrador Sea-Baffin Bay, Greenland Sea and Barents-Kara Seas), the net flux is close to zero.



How can we reconcile this deficit of ice melt with the fact that 2007 was a sea ice low, with a large part of the ice concentration loss concentrated in the Pacific sectors? The explanation resides in the fact that a sea ice extent anomaly is not necessarily accompanied by a sea ice volume or mass anomaly. Indeed, the ice redistribution through dynamical processes from the Chukchi-Bering Seas and Siberian Seas sectors into the Beaufort Sea and Central Arctic sectors has led to a diminution of the ice extent without significantly altering the volume. In the model, this has led to convergence and ridging of the ice, as

hinted by the positive ice thickness anomalies along the Canadian Arctic Archipelago and shown by hotspots of anomalous ice mass import (supplementary materials, Figure S2). September 2007 is not a clear ice volume low in the model (supplementary materials, Figure S1), and whether it is one in summer observations is not obvious (Kwok, 2018; Soriot et al., 2024). It is worth noting that the model simulates an increase in basal melt in the Chukchi Sea in July 2007 (supplementary materials, Figure S2.m), consistent with suggestions that the oceanic heat transport through the Bering Strait increased that summer

(Perovich et al., 2008; Woodgate et al., 2010), but this increase is not sufficient to compensate the export of ice in the budget and it is not a main driver of the ice low, according to the model.

The Arctic 2012 sea ice low shows some similarities, but also differences, with the 2007 one (Figure 4.b). The strongest negative thickness anomalies are centred in the Atlantic side, northward of the Barents and Kara Seas, as well as in the Siberian Seas, consistent with observations. Some ice loss is also visible in the southern fringes of the Beaufort Sea, as in

2007. The Chukchi Sea is experiencing positive ice thickness anomalies, contrarily to 2007. When diving into the ice mass budget, the net flux for the total Arctic is zero. This is again relatively surprising, but this aggregated result hides an important spatial variability. As in 2007, the Barents-Kara Seas sector exhibits a strong positive mass flux anomaly, related here again to preconditioning, with very low sea ice extent occurring in February and March 2012 (supplementary materials, Figure S1). The small positive anomaly in the Labrador Sea-Baffin Bay sector is due to similar reasons, but the positive anomaly in the

Greenland Sea sector is different, as it is rather due to an increased ice import into the sector through Fram Strait. This ice is then constrained along the East Greenland shore, which leads a positive ice thickness anomaly in August. But the most important differences between 2007 and 2012 occur in the higher Arctic encompassing the Central Arctic, Beaufort Sea, Chukchi-Bering Seas, and Siberians Seas sectors, where strong negative mass flux anomalies indicate that the ice actually melted there during the summer 2012. Except in the Siberian Seas sector where the negative anomaly is due to export of ice out of the sector

and therefore compensated by reduced surface melt due to the lack of ice to melt, the negative anomaly in the other sectors (Beaufort Sea, Chukchi-Bering Seas and Central Arctic) is driven primarily by enhanced basal melt, with important influence of surface melt also occurring in the Central Arctic sector. Thermodynamic processes were therefore important in driving the 2012 sea ice low.

As mentioned earlier, an Arctic summer cyclone in early August has been suggested to have been the dominant driver for

the 2012 sea ice low (Parkinson and Comiso, 2013). Yet, no significant influence of the cyclone is visible in the model outputs (not shown). This would suggest that this summer storm was not a determining factor in the sea ice low, supporting the results of Guemas et al. (2013) and Zhang et al. (2013). The cyclone could have had some indirect influence though, through vertical mixing in the Beaufort Gyre. Indeed, the model simulates a strong positive anomaly in the heat content of the upper 100 m of the Beaufort Gyre in 2012 (supplementary materials, Figure S3). This anomaly matches in-situ observations and has been





linked to solar heating of water masses north of the Chukchi Sea that then propagate into the Beaufort Gyre (Timmermans, 2015; Timmermans et al., 2018). The model properly simulates this long-term accumulation of heat, and the strongest anomaly is located in sectors where increased basal melt is most important. Strong winds are likely to have contributed to ice break-up and vertical mixing, bringing this subsurface heat in contact with sea ice. So, the cyclone of 2012 may have had an indirect effect on the Arctic conditions after 2012 but likely not on the sea ice low itself.

To summarise, the summer 2012 sea ice low is related to preconditioning in the Barents-Kara Seas sector and to thermodynamic processes in the higher Arctic, including the presence of a positive heat content anomaly in the Beaufort Gyre leading to increased basal melt. The influence of the cyclone is limited and might rather be indirect. It is therefore a low not only in sea ice extent but also in volume, in contrast with summer 2007, which was a sea ice low in extent, but not in volume. Preconditioning also played a role in the subarctic seas in 2007, but most of the ice extent loss in the higher Arctic was rather related to dynamic

processes leading to convergence and ridging, with some thermodynamic influence through surface melt in the Central Arctic sector.

## 4.2    Comparing two different hemispheres: Arctic 2012 and Antarctic 2022

We now aim to apply the same analytical framework to intercompare summer sea ice lows across hemispheres. The Southern Ocean has experienced a significant sea ice low in 2022, ten years after the Arctic 2012 low, and this sea ice low is well resolved

by the model (Figure 1). The sea ice thickness anomaly is mostly located in the Ross Sea, with negative thickness anomalies of a few tens of centimetres (Figure 5.a, map). Thinning of the ice is also visible in the western part of the King Håkon VII sector (close to the Weddell Sea) and the eastern part of the East Antarctic sector (close to the Ross Sea). The Weddell Sea and Bellingshausen Sea sectors are specific, showcasing both thinning and thickening of the ice. In particular, strong thinning is visible along the eastern side of the Antarctic Peninsula, next to a fairly strong thickening of the ice in the southern part of the

Weddell Sea. These spatial anomalies match well with satellite observations of sea ice concentration anomalies (Turner et al., 2022; Wang et al., 2022) and give here again confidence that the model is skilful and can properly simulate this sea ice low. The ice mass budget anomalies follow a similar spatial pattern (Figure 5.a, insets). For the whole Antarctic region, the net mass flux is negative in 2022, and this negative anomaly is caused by enhanced basal melt but it is partially compensated by enhanced snow-to-ice conversion (which, as we have mentioned, tends to be larger when basal melt is enhanced). This negative anomaly

is evenly distributed between the Ross-Amundsen Seas, King Håkon VII, and East Antarctic sectors; the Bellingshausen Sea sector brings no contribution to the overall mass flux while the Weddell Sea sector contributes with a positive anomaly (less melt than climatological conditions). The lack of melt in the Weddell Sea sector occurs both at basal and surface and is mostly due to preconditioning, with already low ice extent in September (supplementary materials, Figure S4), as documented for the Antarctic as a whole in spring 2021 (Raphael and Handcock, 2022). When looking at the negative anomalies, the East

Antarctic sector differs from the others since the increased mass loss is due to export of ice out of the sector, mostly towards the Ross Sea but also partially towards the King Håkon VII sector. Dynamic processes are therefore driving the ice mass loss there. In contrast, the Ross-Amundsen Seas and King Håkon VII sectors experience ice loss due to strong basal melt, partially compensated by snow-to-ice conversion, a corollary of the lowered freeboard induced by this basal melt. This is also





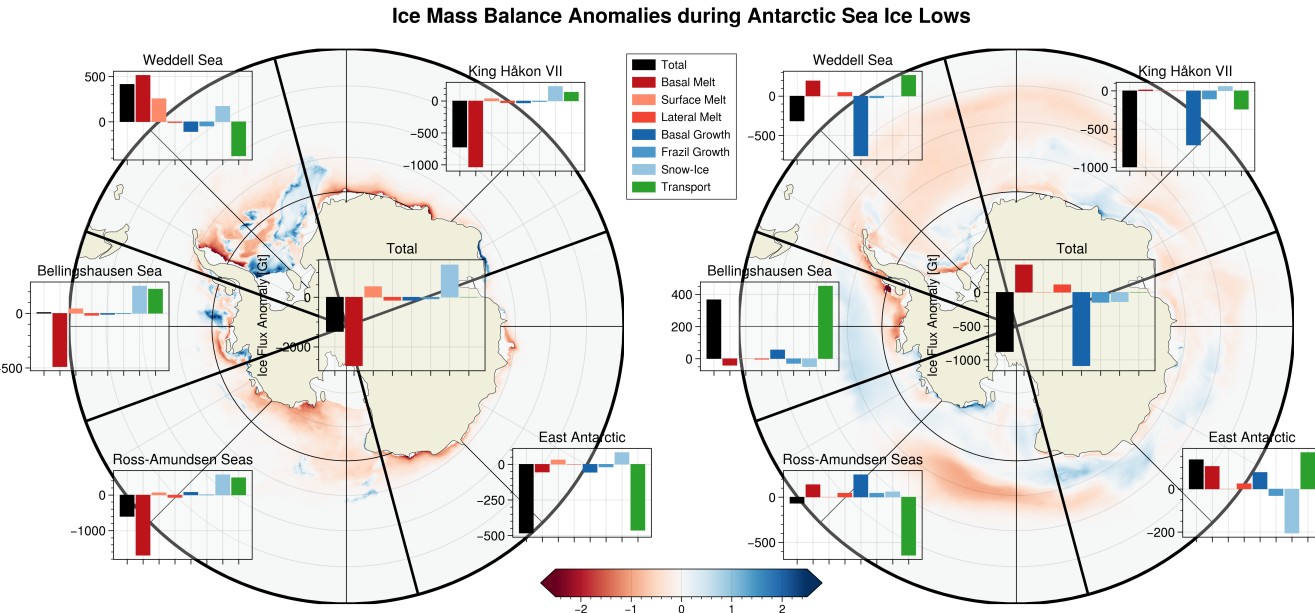

**Figure 5.** Antarctic (a) austral summer 2022 and (b) austral winter 2023 sea ice lows. Spatial distribution for (a) February and (b) September ice thickness anomalies with respect to 1979-2008 climatology (background map) and ice mass budget anomalies integrated from (a) October to February and (b) April to September for each sector (as delimited by black lines) and for the total Antarctic (inset panels; the scales of the y-axes differ among panels). A positive anomaly means an anomalous gain of ice mass, i.e. less ice melt or more ice growth than climatological conditions.

the case in the Bellingshausen Sea sector, except that the snow-to-ice conversion and the import of ice from the Amundsen Sea

compensate entirely the enhanced basal melt. In the Ross Sea, detailed spatial anomalies indicate that equatorward export of sea ice towards warmer waters leads to the increased basal melt offshore (supplementary materials, Figure S5). This northward transport of sea ice has been explained by a deepened Amundsen Sea Low (Wang et al., 2022; Mezzina et al., 2024).

It is interesting to note the overall parallelism between this Antarctic 2022 low and the Arctic 2012 low, where the spatial heterogeneity of the mass budget can be explained by a mixture of preconditioning in some sectors and increased basal melt

in others. Thermodynamic processes dominate in both cases, though dynamic redistribution of ice is also important in the case of the Antarctic 2022 low and can lead to thermodynamic anomalies by transport of ice towards warmer waters. This dynamic redistribution is rendered easier in the Southern Hemisphere due to the geographic configuration that allows for freer movement of ice between sectors as defined here and latitude-wise (see also Maksym, 2019). The main differences between both lows, specifically the respective influences of surface melt in the Arctic and snow-to-ice conversion in the Antarctic, are reflective of

the climatological differences between both regions.



### 4.3 Comparing two different seasons: Antarctic 2022 and Antarctic 2023

The comparison between two hemispheres has shown that dominating processes can be similar in both regions. It can also be instructive to compare two seasons, to see if winter drivers of sea ice minima offer common characteristics to summer minima. In austral winter 2023, the Antarctic sea ice extent was the lowest over the satellite record, five standard deviations below the

climatological average (Espinosa et al., 2024). This is also the case in the model, which simulates a record low sea ice extent of 17 millions km$^2$. This sea ice low is also an ice volume minimum in the model, with negative ice thickness anomalies visible in all sectors, but most prevalent nearshore in the Bellingshausen Sea and offshore in Weddell Sea and Ross-Amundsen Seas sectors. Positive thickness anomalies are also visible in nearly all sectors, closer to the continent, but offshore as well in the Amundsen Sea and in the East Antarctic sector. This distribution of thickness anomalies matches well with observations

(Ionita, 2024, their Figure 2). The mass budget provides here again a decomposition of the processes leading to this ice low. For the Antarctic as a whole, the net mass flux exhibits a strong negative anomaly, which is explained by a negative basal growth anomaly, meaning a strong lack of ice formation. Reduced frazil ice and snow-to-ice conversion are also participating to this negative anomaly, while a non-negligible positive basal melt anomaly (*i.e.*, a lack of melt) partially compensates the negative anomalies. This lack of basal melt might seem surprising in winter, as it implies that there should be basal melt in

winter in a climatological year. This is indeed the case, as Antarctic sea ice is exported equatorward to warmer latitudes in winter, where basal and lateral melt would then occur. The positive basal melt anomaly can then be explained by the lack of sea ice in winter 2023, which would reduce the potential for sea ice export towards warmer latitudes and therefore the associated melt fluxes, compared to climatological conditions. According to the sectorial decomposition, most of the negative mass flux is coming from the King Håkon VII and Weddell Sea sectors, while the Ross-Amundsen Seas sector plays a negligible role and

the Bellingshausen Sea and East Antarctic sectors experience a positive mass flux. A lack of basal growth explains the reduced ice mass in the King Håkon VII and Weddell Sea sectors, along with a small lack of basal melt occurring in the Weddell Sea sector and some anomalous ice export from King Håkon VII sector into Weddell Sea sector. The Ross-Amundsen Seas and Bellingshausen Sea sectors are dominated by transport processes, with ice advected eastward towards the Bellingshausen Sea sector. This leads to younger, thinner ice in the Ross Sea that can then grow faster, leading to the increased basal growth in that

sector. It also reduces the mass of ice that is exported equatorward, and therefore the basal melt there. The East Antarctic sector is more difficult to analyse and understand, though the anomalies there are small compared to other sectors. A small positive mass flux anomaly is driven by a mix of lack of basal melt, increased basal growth, and ice import, partially compensated by a lack of snow-to-ice conversion. But this sectorial budget aggregates a spatial heterogeneity within the sector, with a different behaviour between the western region close the Kerguelen Plateau (west of Totten Glacier, around 110 °E) and the

Adélie coast (east of Totten Glacier). The ice import anomaly is driven by westward ice transport from the Ross Sea. More transport redistributes ice offshore of the Adélie coast, instead of westward along the shore, to a location where basal melt typically occurs even in climatological winter. This basal melt is therefore reduced in winter 2023, leading to the concomitant anomalous lack of snow-to-ice conversion (supplementary materials, Figure S6). In that area, but also in the western part of





the sector, the nearshore experiences increased basal growth, concomitant with offshore export of the ice that allows for more

ice growth. Despite those considerations, the anomalies are small and play a negligible role in the winter 2023 sea ice low.

The importance of the basal growth in generating the anomalously low sea ice extent and volume in winter 2023 calls for further investigation of the ocean state. Following the hypothesis from Purich and Doddridge (2023), a positive ocean heat content anomaly at subsurface, between 100 and 200 m depth, would lead to enhanced ice-ocean heat flux and therefore to reduced basal growth or increased basal melt. The model replicates those observed positive anomalies for most of the Southern

Ocean between April and September 2023 (Figure 6, background map). The ocean heat content is here calculated by looking at the departure of the ocean temperature from the freezing temperature (taken as constant at -1.85 °C) scaled by the seawater density and heat capacity ($\rho_0 = 1026$ kg m$^{-3}$ and $c_p = 3992$ J kg$^{-1}$ °C$^{-1}$), and integrated over the layer depth (100 m). The climatological seasonal cycle (over 1979-2008) is removed to obtain spatial anomalies. Overall, the heat content anomalies match spatially with the ice thickness anomalies, suggesting a strong coupling, in the model, between the sub-surface and the

surface. The strongest positive heat content anomalies, located in the Bellingshausen Sea, are superimposed with the strongest ice thickness negative anomalies. Similarly, the nearshore positive ice thickness anomalies around the continent are overlaying negative heat content anomalies. The long-term evolution of the subsurface ocean heat content better highlights how anomalous austral winter was in 2023 (Figure 6, insets). In the Weddell Sea and King Håkon VII sectors, the heat content was at a record high, for all months. Those two sectors are those which experienced strong negative mass flux, due to lack of basal growth.

This is also the case for the whole Antarctic. The Bellingshausen Sea and East Antarctic sectors also show anomalously high ocean heat content, but not record high, and the ice mass there is dominated by other fluxes than basal growth. The Ross-Amundsen Seas sector stands out, as it is the only sector that exhibits a decrease of heat content over the run period for the annual mean, a satisfying match with observational results though considered latitudes are different (Purich and Doddridge, 2023, their Figure 3.c). In the Ross-Amundsen Seas sector, the April heat content is at a record minimum in 2023, but shows a

anomalously reduced heat loss over the ice growth season to exhibit close to record high heat content in September. This could indicate a different causality between ice thickness and heat content and will be further investigated in another study. Except in the Ross-Amundsen Seas sector, the subsurface oceanic heat content is anomalously high in winter 2023 and is likely a dominating driver of the subsequent ice low, although the influence of the atmosphere as a driver of the heat content anomaly or as a medium between the ocean and the ice cannot be ruled out. Nonetheless, this is coherent with a previous study that

demonstrated that most of the winter 2023 sea ice extent anomaly was explained by warm conditions in the Southern Ocean (Espinosa et al., 2024).

In both summer 2022 and winter 2023, the thermodynamic processes drive the sea ice lows, enhancing basal melt in summer 2022 and reducing basal growth in winter 2023. The importance of thermodynamic fluxes, especially at the ice-ocean interface is therefore a common denominator between both Antarctic sea ice lows. As seen above, the ocean heat content is likely

contributing to the winter 2023 sea ice lows. Yet, it is worth mentioning that, at the beginning of the melt season of the austral summer 2022 (*i.e.,* in September 2021), in the Ross-Amundsen Seas sector which experiences the strongest basal melt anomaly, the ocean heat content is relatively low (Figure 6). The King Håkon VII sector, which also experiences a strong negative basal melt anomaly, contains more heat than average, but it is not at record high neither. Only the Bellingshausen Sea sector heat





content is close to record high in September 2021, matching the increased basal melt. Seasonal changes in the stratification

of the upper Southern Ocean would explain the apparent stronger influence of subsurface heat content in winter 2023 than in summer 2022. Indeed, ice melt occurring in spring and summer would strengthen the stratification and better shelter the surface ocean from warmer subsurface waters. Despite those disparities, both sea ice lows occur after the overall ocean heat content increase, visible in the model after 2016 (Figure 6, central panel) and likewise documented in observations (Purich and Doddridge, 2023). Disentangling the causality between this ocean heat content increase and changes in sea ice mass fluxes is

beyond the scope of this study. But, as hinted in the introduction, there is not a single driver of the lows, and preconditioning and dynamic redistribution of the ice between sectors and within each sector are important factors as well.

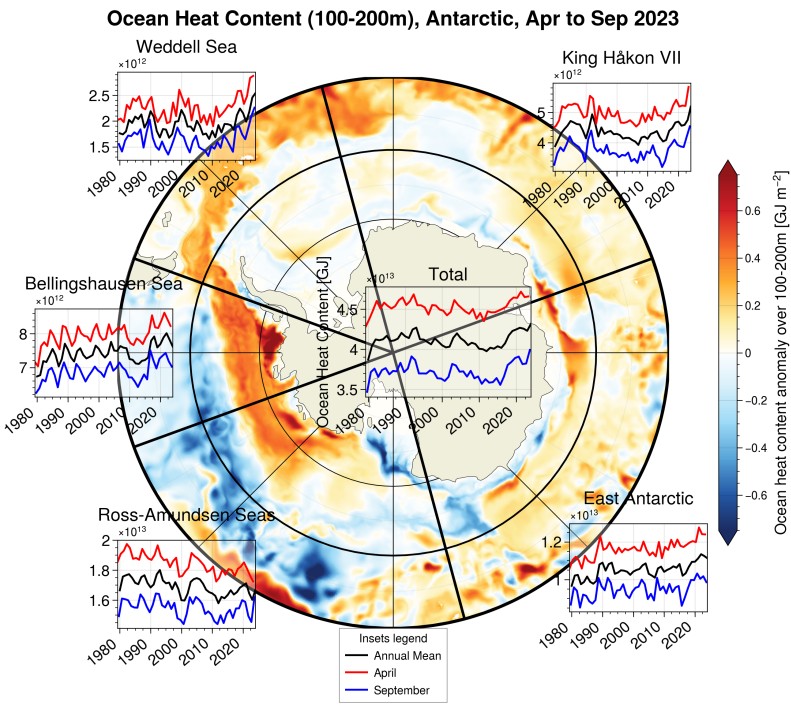

**Figure 6.** Antarctic subsurface ocean heat content anomaly (map) and evolution (insets). The ocean heat content is integrated over the 100-200 m subsurface layer and is calculated with respect to a reference freezing point (-1.85 °C). Anomalies averaged from April to September 2023 highlights oceanic hotspots during the winter 2023 sea ice low (background map). The long-term evolution of heat content for the Total Antarctic and for each sector is also provided (inset panels; the scales of the y-axes differ among panels), calculated for latitudes south of 50 °S (black parallel on the background map), calculated as annual mean (black lines), for the beginning of the ice growth season taken as April (red lines) and for the beginning of the melt season in September (blue lines).





## 5 Discussion and conclusion

The ice mass budget analysis conducted on output from an ocean-sea ice numerical model has highlighted similarities and differences in the mean state and evolution of sea ice cover in the Arctic and Antarctic regions, as well as the causes of several sea ice lows that occurred in recent years. Mass fluxes at the ice-ocean interface dominate the climatological behaviour of sea ice, with basal growth and melt being the dominating fluxes for the ice seasonal cycle, and with basal melt further gaining in importance to the detriment of surface and lateral melt. In regions where surface melt plays a greater role, such as the Beaufort Sea and Central Arctic, this relative decrease in surface melt could lead to a even greater control of the sea ice thickness by the heat balance at the ice-ocean interface. Note that the heat balance at the ice base is not exclusively driven by the ocean and can be strongly influenced by atmospheric processes, as will be discussed further below. Fluxes that do not dominate the climatological seasonal cycle, *i.e.* excluding the basal growth and melt, provide some differences between both poles, with surface melt representing a non-negligible influence in the Arctic, while the higher snow precipitation in the Antarctic reduces surface melt but leads to important snow-to-ice conversion. Those discrepancies are also emerging when comparing sea ice lows. The anomalous sea ice conditions in summer 2007 in the Arctic are due to a mix of preconditioning and ice transport leading to reduced ice extent but stable ice volume in the high Arctic, while conditions in summer 2012 are more clearly due to increased basal melt, with an important role of preconditioning as well. The picture is similar for the Antarctic summer 2022, with preconditioning occurring in the Weddell Sea sector and increased basal melt elsewhere. The Antarctic winter 2023 sea ice low is due to a lack of basal growth. Therefore, fluxes at the ice-ocean interface are again dominating the ice lows, with enhanced basal melt in Arctic summer 2012 and Antarctic summer 2022 lows, and strongly reduced basal growth in Antarctic winter 2023. But the hemispheric differences also show up in those comparisons, since snow-to-ice conversion plays a strong role in partially compensating the ice loss in Antarctic summer 2022, and surface melt significantly increased ice loss in the Central Arctic sector in summer 2012. Those mass fluxes should not hide the importance of two other processes at more local scales: preconditioning and dynamical redistribution of ice. Preconditioning is indeed an important driver of sea ice lows in subpolar seas in Arctic summer 2007 and 2012 and in the Weddell Sea sector in Antarctic summer 2022, leading to anomalous positive fluxes since there is less ice than usual to melt. Ice transport leads to spatial redistribution of sea ice, decreasing the potential for ice melt as in the Siberian Seas sector in summer 2007 and 2012 and in the East Antarctic sector in summer 2022 or increasing that of ice growth as in the Ross-Amundsen Seas sector in winter 2023. But most important, it leads to ridging and compression in Arctic summer 2007, meaning this ice low is a clear low in extent but not as clear in volume, according to our model. Overall, none of the investigated sea ice lows can be boiled down to one cause, as they are rather due to a mix of preconditioning, thermodynamic, and dynamic processes. This is to be expected, as extremes are more likely to be a superposition of factors, in order to reach a state that is statistically less likely.

Our model, like any model, suffers from some weaknesses in the simulation of the mean sea ice state. As mentioned in Section 2, a negative bias in the minimum sea ice extent is clear for both hemispheres, but less so in the maximum sea ice extent. The consequence is an amplified seasonal cycle of sea ice extent in the model, and the late timing of the maximum ice extent leads to a faster and stronger than expected melt season. An obvious question is then: how can this impact our results?





The fast melt season is likely to lead to a positive bias in all melting and growing terms in the climatological seasonal cycle. It could also lead to an underestimation of the relative importance of basal melt in the Arctic, going hand-in-hand with an overestimation of the relative importance of surface melt, due to the fact that surface melt peaks earlier than basal melt in the high Arctic. Less ice would then be available for basal melt when solar radiation decline and surface melt decreases. But, when

investigating the sea ice lows, the biases on the mass fluxes should be reduced, since we only consider anomalies and therefore remove the climatological biases from the fluxes. Moreover, the magnitudes of the ice extent anomalies are reasonably close to observations (a few percents of difference compared to the total seasonal cycle amplitude, Figure 1.b and d). So most of the bias should be in the mean state, rather than in the variability. Nonetheless, the biased mean state of sea ice could also influence anomalies indirectly. For example, during sea ice lows, the lower sea ice extent in the modelled climatology in subpolar Arctic

seas would lead to a closer-to-normal melt in the model than in reality, underestimating the positive anomalies in the melting terms, while overestimating it in sectors of the higher Arctic. It could also lead to decreased ice transport from the Siberian Seas sector to the Central Arctic sector in 2012, since the Siberian Seas sector is close to ice free early on in the melting season in the model but not in observations. In the Antarctic, another issue arises from the negative minimum extent bias: the model cannot exhibit as much interannual variability as in observations, since the mean state is already so low. This would

reduce the magnitude of the anomalies and could explain why the variability is reasonable compared to observations. Despite those obvious shortcomings, the model ice mass flux climatology and trends are in good agreement with the standing scientific knowledge, and the simulated sea ice lows are all very similar in spatial and temporal feature with results from other studies. We are therefore confident that the dominating processes are properly captured, at least qualitatively, and that any discrepancy due to the model bias would be reflected in the magnitude of local terms, but not in the magnitude of the spatially integrated

fluxes, nor in the sign of the anomalies.

For all sea ice lows, except the Arctic summer 2007, the dominating processes leading to the ice mass loss occur at the ice-ocean interface, either due to basal melt or lack of basal growth. This is an interesting and important outcome of the analysis provided here. While atmospheric conditions are often put forward to explain sea ice lows, either through reduced cloud coverage, summer cyclones, or strengthened atmospheric circulation patterns (Olonscheck et al., 2019; Docquier et al.,

2024), our results might seem to rather highlight the oceanic influence. Yet, we cannot directly link these enhanced basal fluxes to a prevalent influence of the ocean. Indeed, a change in basal growth or melt is determined by the heat budget at the ice-ocean interface, which is a balance between the oceanic heat flux and the conductive heat flux within the ice, itself depending on the temperature profile in the sea ice (for more details, see e.g. Maykut and Untersteiner, 1971). The latter depends on the ice surface temperature, determined by a balance between the atmospheric heat fluxes and radiations. The temperature ice

profile also depends on solar radiation, which can penetrate through the ice and warm either some intermediate ice layers or directly the under-ice ocean layer if the ice is thin enough. A modified ice temperature profile will then impact the heat balance at the base of sea ice, and the resulting basal mass fluxes. Therefore, changes occurring only in the atmosphere could still have an impact on basal mass fluxes, without inducing any surface melt if the ice surface temperature remains below the melting point. We therefore cannot rule out the direct influence of atmospheric forcing in the ice melt, despite the fact that

most of the mass flux anomalies occur at the ice-ocean interface. Moreover, the ice-ocean system is a strongly coupled system.



Changes in atmospheric properties can have important though indirect consequences on the ocean state. An obvious example is the ice albedo feedback, that can originate from increased insolation as suspected in summer 2007, and lead to warming of the upper ocean, inducing more ice melt at the ice-ocean interface if water masses are advected towards the marginal ice zone (Schweiger et al., 2008; Woodgate et al., 2010). At wider spatial scales, the oceanic heat transport can also be related to 550 atmospheric conditions (Docquier and Koenigk, 2021). Despite those considerations, a number of recent studies have shown the importance of oceanic heat transport to explain ice melt (Docquier et al., 2021; Aylmer et al., 2022, 2024). In the case of the analysed sea ice lows, the long-term positive trend of the Beaufort Gyre heat content is a well documented feature (Timmermans et al., 2018) and is realistically reproduced by the model (supplementary materials, Figure S3). This increase is a likely source of the observed increased basal melt seen in the Arctic summer 2012, and it is well superimposed with a 555 dislocation of the ice cover. Similarly, for the Antarctic, the presence of strong positive heat content anomalies in winter 2023 in sectors that experienced the least basal growth is a good indicator of the importance of subsurface heat content anomalies in modifying the ice cover. Furthermore, the widespread increase in subsurface ocean heat content after 2016 visible in most sectors (Bellingshausen Sea, King Håkon VII, a couple years later in Weddell Sea and East Antarctic) matches well with observations and is a good indicator that some important changes are at play at the scale of the Southern Ocean. The origin of 560 this subsurface warming is still debated (Purich and Doddridge, 2023) but could be due to some ice-ocean feedback involving brine-related changes in the stratification (Goosse and Zunz, 2014).

The dominance of mass fluxes at the ice-ocean interface is not at odds with our current scientific understanding. Observations tend to highlight a large diversity of processes at the sea ice base and a strong influence of basal melt (e.g. Perovich et al., 2008), but the lack of in-situ observations prevents any quantitative validation of the model. This calls for a need of year-round 565 observations of ice-ocean fluxes, including heat fluxes, as well as for under-ice properties as close to the ice-ocean boundary layer as feasible. The Antarctic InSync program, part of the Ocean Decade effort, and the incoming 5[th] International Polar Year (2032-2033) campaigns would provide a great framework for improving our understanding of processes at the ice-ocean interface. Emerging observational platforms such as the seasonal ice mass balance buoy (Jackson et al., 2013; Planck et al., 2019) or the ice tethered profiler (Toole et al., 2011) and its recent evolution as an ocean tethered profiler (O'Brien et al., 2023) 570 provide the technological means to observe crucial properties of the ice-ocean system. A more widespread and systematic deployment of those autonomous platforms would be invaluable for model development and validation.

*Author contributions.* BR, FM and TF outlined the study. AB performed the simulations. BR made the analyses and the figures, and all the co-authors contributed to the interpretation of the results. BR wrote the manuscript with inputs from all co-authors.

*Competing interests.* None of the authors have any competing interests.



*Acknowledgements.* This study is supported by the Belgian Science Policy Office (BELSPO) under the RESIST project (contract no. RT/23/RESIST). DD is funded by BELSPO through the RESIST project. The present research benefited from computational resources made available on Lucia, the Tier-1 supercomputer of the Walloon Region, infrastructure funded by the Walloon Region under the grant agreement n°1910247. We acknowledge EuroCC Belgium for awarding this project access to the LUMI supercomputer, owned by the EuroHPC Joint Undertaking, hosted by CSC (Finland) and the LUMI consortium through EuroCC Belgium."



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
