# Peer review of "Anatomy of Arctic and Antarctic sea ice lows in an ocean–sea ice model"

_EGUsphere, 2025_

## Referee Comment (RC2)

**Richaud et al.: Anatomy of Arctic and Antarctic sea ice lows in an ocean–sea ice model**

The authors in this study use the NEMO-SI3 model to investigate extreme lows in sea ice extent in both hemispheres over the historical period by using sea ice mass budget terms, both hemispherically and regionally. These modeled mass budget terms are particularly valuable as they are not available in observational products but they provide insight into the processes driving the low extent years. The authors find that basal processes (melt and growth) become increasingly dominant in the mass budget in both hemispheres. Additionally, they find that during case study low extent events the importance of thermodynamics and dynamics can vary, as can the importance of difference processes by season or hemisphere due to the local ice state and drivers. Overall, this study is well designed and provides insight into how sea ice evolution is changing. I have some major concerns, detailed below, about the analysis that I would recommend being addressed in a major revision before publication.

**Mass budget analysis**: The authors present their mass budget analysis in section 4 as absolute mass loss anomalies, which itself is dependent on mean state of the sea ice. As a result, I found that the conclusions and graphs were confusing and a bit misleading about the results and more care needs to be taken with how this is presented. In years with extreme low extent anomalies the anomalously high positive ice mass budgets were sometimes explained as "preconditioning" based on the mean state. I recommend showing the anomalous mass budget terms normalized by the total ice area each year to better understand how the budget terms compare to other years. Additionally, it would be beneficial to show how the dominant budget terms change over time to support your assertions about basal melt change (e.g. Line 480-485) since Table 1 just gives a single linear trend but doesn't show the budget terms over time. It would also be helpful to show a timeseries of the fractional contributions of each mass budget term, like this figure (right) from Bonan et al. 2021 (https://doi.org/10.1088/1748-9326/abe0ec), to show how basal terms become more dominant. Again, you'd want to normalize this so that it isn't just reflecting a mean state change of less melt due to less volume.

[Figure]

**Ocean Heat Content:** OHC is mentioned several times (Line 353, 358, 455, 454, 466) but is not clearly shown in figures and needs more analysis to support your assertions. For the Arctic cyclone in 2012, you should have a figure like Fig.6 that shows the OHC aspect of this story and the impact on basal melt. For the Antarctic, you should do a spatial correlation of OHC and ice anomalies to quantify the relationship described at Line 445. Since you're analyzing winter months, there may be important implication of OHC anomalies that are not just from surface heating or the albedo effect. Additionally,

for the panels in Figure 6, you say you are integrating OHC south of 50S. In many of the sectors this means you're including anomalies that are not actually in contact with the sea ice since they're well north of the seasonal sea ice zone. This makes your analysis at Lines 454-456 and 466-467 of the anomalously low OHC in the Ross-Amundsen sectors confusing since much of the anomalously low areas are not in contact with sea ice. I recommend you mask Figure 6 to show only that year's sea ice zone. Additionally, you may want to calculate a regional mean OHC anomaly only over points that are within the SIZ to better understand how the sea ice will be affected by OHC year to year.

**Antarctic transport:** The discussion from Lines 229-241 on Antarctic transport was confusing. You mention the "circumpolar current", which probably refers to the ACC that is on the northern edge of the sea ice zone. What about transport from other currents like coastal currents which are in the reverse direction? I think that the transport piece itself is complex and needs to be more carefully assessed and discussed.

**Clarity:** Much of the paper consists of VERY long paragraphs that are difficult to follow for a reader and often contain several points (e.g. Lines 132-162, 368-393, 402-435, section 5). Please break these paragraphs up so that they're clearer and more honed to the points you are making. A few places I found particularly challenging to follow were Section 4 and the discussion conclusions. Section 4 is framed as case studies of different hemispheres and years. I think it could be more powerful to frame these in terms of processes. For example, for section 4.1 a title like "the impact of ice volume on extremes" will help underscore the conclusion is that sea ice extent alone does not tell the full story and volume is necessary. Section 4.2 seems to be comparing hemispheres, but depends on how free drift and dynamics vs. thermodynamics impacts the extremes. Section 4.3 is better in that it clarifies seasonal differences are the focus. You may want to compare seasons for the Arctic too to confirm your results and see if the processes driving each hemisphere differ similarly by season. The discussion and conclusions sections are wordy with long paragraphs that make it difficult to separate the primary conclusions from the discussion. Perhaps make conclusions its own, small final section and better hone the paragraphs in this entire section.

**Specific figure comments**:
- Figure 1: Since a big part of this paper is showing that sea ice volume or mass is critical for understanding extreme ice extent loss events (e.g. line 166, 322, 326, etc.), you should include panels with the timeseries of volume anomalies in this figure. These are already Figures S1a and S4a, but they should be included as main figure panels as well. Additionally, it could be beneficial to compare to a hindcast that uses data assimilation (e.g. PIOMAS for the Arctic, SOSE for the Antarctic?) to see how well your model compares to some estimates of these variables.
- Figure 1: Panels a and c are averages over some years – you should list the years they are averaged over in the figure caption and show the standard deviation to indicate variability.

- Figure 3: This is a great figure! The map legend doesn't show white on white for the contour, so this should be fixed.
- Figures 3/4/5/6: These figures are nice, but the graphs overlain on the maps makes them hard to read, especially the center Antarctic one. Additionally, there are no y axis labels on the regional graphs. For figures 4/5/6, the bar charts axes limits aren't consistent, so it is hard to compare the panels with one another. Consider have a white, opaque background or reorganize the figures so that the graphs are separated from the maps and easier to compare, and please be sure to label axes.

**Minor comments**:
- Line 27: You should also cite the following two Holland papers about variability in a changing climate: https://doi.org/10.1007/s00382-010-0792-4 (Figure 8), https://doi.org/10.1029/180GM10 (Figure 3c, not plate 3).
- Line 52: You should mention the 2016 low was after years of a small positive trend.
- Line 120: Does the floe size (that impacts lateral melting) change or is it constant?
- Line 137: "millions" should be "million"
- Line 152: why are you considering just the last two decades? Is this justified?
- Line 167: it isn't true that thickness "cannot" be observed. Rephrase this to say: "However, it cannot be compared at similar spatial and temporal scales as areal extent since there are not long-term observations of thickness."
- Line 220: this sentence was not clear. Maybe you mean the signs of the transport are consistent with transpolar drift, but the sentence needs to be clarified.
- Line 248-251: This is consistent with Bitz et al. 2005 (https://doi.org/ 10.1175/JCLI3428.1) who show that the Antarctic ice edge is determined by ocean heat and in the Arctic the ocean mainly impacts the Atlantic sector.
- Line 257: "the Antarctic"
- Line 304: "more dynamical" than where? The areas you are referencing against isn't clear and this could be expanded.
- Line 327: I don't think "not obvious" is the right wording. Do you mean there aren't observations to show this? Could satellites ice thickness earlier in the year (e.g. March) help determine the ice state at the start of the melt season?
- Line 331: "ice low" I believe means extent only, but please clarify since a major conclusion is that ice volume lows matter too.
- Line 334: Clarify "ice loss" means negative extent negative anomaly only.
- Line 335: change "contrarily to" to "unlike in"
- Line 362-363: Please make clearer that ice volume vs. ice extent anomalies are not always linked as the key point of this section's analysis. This should be highlighted more as the topic sentence and then explain how 2012 is thermodynamic but 2007 is dynamic for high Arctic.
- Line 373: "sectors are specific"- what does this mean? Maybe better wording would be "sectors exhibit mixed anomalies"
- Line 407: remove "visible"
- Line 566: this final paragraph has a very awkward transition to justifying observational campaigns instead of reiterating this study's findings or explaining how those campaigns (which are limited in temporal and spatial scope) could contribute to understanding hemispheric mass/volume budgets and extrema.

---

## Referee Comment (RC3)

**Review of: Anatomy of Arctic and Antarctic sea ice lows in an ocean-sea ice model**

by Benjamin Richaud, François Massonnet, Thierry Fichefet, Dániel Topál,
Antoine Barthélemy, and David Docquier

July 29, 2025

**Manuscript Synopsis**

This manuscript, using an atmosphere forced ocean-ice historical reconstruction attempts to analyze sea ice flux mass contributions for exceptional sea ice events relative to climatology. The paper uses the common methodology to compare and contrast sea ice events in the Arctic and Antarctic, as well as comparing/contrasting melt and freeze-up season events. The paper could provide a useful measure examining exceptional sea ice events, but suffers considerably by confusing and incomplete graphics as outlined in my major comments below. In particular, the pre-existing sea ice anomaly plays a huge role in allotment of mass balance fluxes into anomalous fluxes, particularly for melt events, where there is a definitive upper bound on sea ice removal (you cannot remove more ice than what already exists). The authors discuss this *"pre-conditioning"* (their term) in the text, but the lack of a graphical representation of this term in their budget can lead to confusing interpretation of the results, particularly by a reader seeking quick visual summarization of the results.

**My recommendation is Major Revisions**

**Major Comments**

1. Fig. 4 & 5. I found the presentation of these figures very hard to follow, as the discussion relies heavily on an additional term (preconditioning or initial mass anomaly) that requires careful reading of the manuscript to draw out. As the figures stand now, it is very easy to convince oneself that positive flux anomalies mean an decrease in sea ice mass, when in actual fact they mean the the opposite (increase in sea ice mass), but in a vast proportion of the sea ice thickness anomalies, this incorrect assumption does seem to visually confirm – and in some cases (Labrador Sea / Baffin Bay) seems to be wholly nonsensical. To make graphical interpretation much simpler: .

   - The sea ice thickness anomaly **must** be over the **same** period (May to August) as the flux anomalies (and not just August).

   - This still does not lead to a visual flux closure as it does not account for the *initial* sea ice mass anomaly (pre-conditioning in the author's terminology), which in many, if not all cases is the main offsetting factor. Therefore an additional "preconditioning" pseudo-flux should be added to the bar chart representing the initial mass anomaly (technically it should be for 30 April, but an average over April should be close enough if more convenient/smoother). I initially though you would need to convert this into a flux – but if I understand correctly the fluxes are already time integrated into mass gain over the 4 months? [You would likely not wish to add this pseudo-flux to the total, just leave it separate.]

   - Only then will it the figure visually balance the fluxes in the sector with the mass loss/gain contours.

- It will also visually confirm large segments of the text which discuss that the apparent flux anomaly is actually due to "pre-conditioning" (i.e. the initial mass anomaly), with the *increase* in anomalous ice mass fluxes (i.e. mass growth) being *largely* offset by the initial anomaly. In other words, there is an increase in anomalous ice growth largely due to there being less ice than climatology to melt!

- Ultimately, the usage of anomalous fluxes seems to be less than informative, the size of the flux ultimately being hugely dependent on the underlying sea ice volume. A better strategy (with no guarantee of success) might be to use normalized (either by total ice volume, or ice volume change, the latter assuming a definitive melt/freeze sign by sector) fractional flux anomalies. For instance does the fraction of basal sea ice melt increase or decrease from climatology in the exceptional years? Note: The fractional flux could be greater than 1, or less than zero. Sign conventions, for lack of better terminology, might be messy. I do not suggest pivoting to such an analysis now, I would view this manuscript as a learning process in best practices in this regard.

- Examples of confusing aspects:

  (a) Erroreous statement of Major Comment #3.
  (b) Statement concluding Subsection 4.1 (Minor Comment #1)
  (c) Using text explanations to highlight effect of pre-existing mass anomaly *(preconditioning)* without additional graphical assistance (ll. 311, 314, 338, 360, 363, 383, 394, 489–505).
  (d) Large sea ice growth flux anomaly in Baffin and Hudson Bays/Labrador Sea sector with only a small manifestation of sea ice loss in the Canadian Archipelago.

2. Ocean Heat Content: I am not entirely convinced of all the claims made in the manuscript with regards to increased heat content leading to increased basal melt.

  (a) The stated alignment of the increased heat content (Figure 6; red; numerically positive) and decreased sea ice volume (Figure 5b; red, numerically negative) do not line up as well as suggested as demonstrated in the enclosed animated gif which purposes to overlay the two (I see a lot of alternating red/blue). Caveat: As with my comments with regards to Figures 4 & 5, the heat content (Figure 6; April to September) does not align in time perfectly with the sea ice volume (Figure 5b; September) either.

  (b) The choice of the 100-200m heat content is a little confusing, and not justified. The winter time mixed layer depths [Uotila et al., 2019] range from 100 to 300m, which means an increase or decrease in the mixed layer could have opposing tendencies in the top 100m and 100-200m – if the mixed layer increases one might expect the upper layer to warm while the lower layer cools (increased surface mixing with the warmer below mixed layer waters), with the opposite cooler surface, warmer 100-200m if the mixed layer decreases (isolates the surface).

  (c) The previous point is very well illustrated in Figure 2 of the reference Zhang et al. [2022]. Indeed, the zero lag in that paper would seem to require an accompanying negative anomaly in 100-200m heat content. The mechanism also requires a long period lagged relationship that I see no evidence of here.

  (d) I might speculate that the heat contents are sea ice driven: Lower sea ice creation implies lower brine rejection and increased stratification (isolation) of the surface waters, increasing the 100-200m heat content. This would explain the relative uptick in Antarctic September heat content relative to April heat content seen at the end of the pan-Antarctic time series in Figure 6 – but it is also difficult to see if this is an isolated event, or a common occurrence.

  (e) As the authors state, the increased/decreased basal melt/growth may be driven by the atmospheric forcing, especially in low sea ice thickness states as the downward heat fluxes directly heat the ocean surface layers.

  (f) I do not advocate that my speculations, or any of the alternative explanations are more or less likely than the mechanisms suggested by the authors. I do suggest there is a lack of current evidence in the manuscript for any conclusions connecting the heat content to the loss of sea ice volume. (Seasonal) Lead/lag relationships may be critical.

3. Erroneous statement: l. 328. The manuscript states there is an increase in basal melt in the Chukchi Sea in July 2007. Supplementary figure S2m-o shows a blue (positive) basal melt anomaly in the Chukchi Sea. But positive flux anomalies are defined as anomalous gain of ice mass. Therefore this is not an increase in basal melt, but a decrease in basal melt. If I am incorrect, please correct me, but this does demonstrate my confusion generated by the figures. I suspect this positive basal melt anomaly is completely due to *"preconditioning,"* – i.e. there is a anomalous lack of sea ice to melt.

   - Similarly, the Chukchi and Bering Seas sector shows a net positive basal melt flux (so again decreased basal melt) in the bar charts of Figure 4a.

   - ll. 329. If I am not confused, and the basal sea ice melt is actually decreased, the connection to ocean heat transport may no longer be appropriate, however, you should have stated (the perhaps obvious, nevertheless still useful) that there are observations of increased **northward** heat transport. I briefly contemplated the authors meant there was an observed *southward* transport of heat to match the flux anomaly.

   - The statement "this increase (in sea ice mass) is not sufficient to compensate the export of ice" is correct – but again only added to my level of confusion.

   - Please check your characterization of your flux sign convention elsewhere in the manuscript. Having noticed this, I cannot convince myself there may be other instances where I have matched my interpretation of the sign convention in the graphics to match the text commentary (i.e. I can be easily confused into agreement).

**Minor Comments**

1. ll 362-363: "It (2012) is therefore a low not only in sea ice extent, but also in volume, in contrast with summer 2007." This comment **cannot** be made here without specifying you are excluding the seasonally ice covered Labrador Sea / Baffin Bay and Greenland Sea and Barents-Kara Seas sectors as previously mentioned in the text. Readers just reading the section concluding remarks (it does happen) will immediately refer to figure 4 and both conclude you have this backward – 2012 has no change in volume, and 2007 has a low in sea ice volume (i.e. invert your sign convention).

   - But if you also include the pre-conditioning flux this will also be rendered visually correct.

2. ll. 81-82. There are considerably more examples and research concerning climatic implications of changes to sea ice [e.g. Screen, 2013] – I would normally provide a more extensive list, but I am stressed for time here (no conflicts in solitary suggestion).

3. ll. 110-111. The tri-polar grid is designed to remain eddy-permitting in the Arctic (grid cells of order ~12km). I should probably know this, but even so, others readers might not. Does the eORCA025 grid remain eddy-permitting throughout the Antarctic domain?

4. l. 117. Is it standard to have equal numbers of sea ice and snow layers (2+2)? I obviously do not know, but I seem to recall the multi-layer thermodynamics sea ice models I have dealt with have more sea ice layers than snow layers. Is there a rationale for this?

5. l. 151: *entire time series*. I assume this is your entire *analysis* time period (1979-2023), but perhaps it is worth repeating here? And the last two decades are presumably 2004-2023?

6. Figure 1b/d: From the looks of the plot, I assume the numerical minimum for both the Arctic and the Antarctic has a time value assigned by *exact* time in year (year + month + day), or in other words, the Arctic sea ice minimum for 2009 is more closely aligned (2009.75, or slightly to the left) of the 2010 grid line than the 2010 (2010.75) minimum. The same applies to the Antarctic, but in this case it is more closely aligned with the correct calendar year. If so (or if not) this should be communicated in the caption. The same question can apply to Figure 6 – with not as much consequence – are the annual mean (+0.5) aligned in time with April (+0.25) and September (+0.75), or are they offset by 0.25? Similarly figures S1 and S4.

7. l. 171. Units are slightly confusing, suggest reordering somewhat (in kg for sector analysis, and kg m$^{-2}$ for individual grid points). However, it is also important this information be added to the figure captions – at least in the first instance of occurrence in a figure. (Figure 3, 4, 5 for sector flux values; Already included for grid values (S2, S3, S5, S6).

8. l. 192. Keen et al. [2021] is an extension of Keen and Blockley [2018] to a multi-model analysis. I would think that (no conflicting interest) the original budget analysis would be a more appropriate citation. Keen et al. [2021] would remain applicable for placing this manuscript's results within the CMIP context (ll. 224, 226, 289, 290). Note the DOI is correct (`https://doi.org/10.5194/tc-15-951-2021`, but the link does not work properly (across two lines, with the automatic line numbering interfering?) for Keen et al. [2021].

9. The sector Labrador Sea and Baffin Bay also includes Hudson Bay, which is likely an equal contributor to the sea ice mass changes over the May-August period. While Labrador Sea / Baffin and Hudson Bays is likely too lengthy for labelling purposes, I favour the more accurate East Canada Arctic moniker (Eastern Canadian Arctic is more grammatically correct, but longer).

**Grammar and Typographical Errors**

1. ll. 71-75. Mixing explicit time specific explanations (anticyclonic flow in 2007, summer storm in 2012, ...) with more generic explanations (anomalous ocean heat inflow, subsurface conditions, ...). The paragraph would likely read better if you separated this lengthy sentence into time specific explanations and generic explanations.

2. l 280. an hemispheric → a hemispheric (as correct on l.174) (ChatGPT: *Correct: an hemispheric level*)

3. l 469. but it is not at record high neither → but it is not at a record high either

   - Personally, I would refrain from using "record high." This implies more information than is known. I would recommend "but it is not at the highest point in the modelled timeseries either. Only Bellingshausen Sea sector heat content is close to its peak value.
   - It is also very difficult to precisely locate 2021 on the displayed timeseries. I assume this peak is confirmed at 2021 in the numerical data?

4. l. 517. a few percents of difference → a few percent of difference

5. l. 539. radiations → radiation (non-countable quantity; ChatGPT: *Correct: There are many types of radiations.*)

**References**

A. Keen and E. Blockley. Investigating future changes in the volume budget of the Arctic sea ice in a coupled climate model. *The Cryosphere*, 12(9):2855–2868, 2018. doi: 10.5194/tc-12-2855-2018. URL `https://tc.copernicus.org/articles/12/2855/2018/`.

A. Keen, E. Blockley, D. A. Bailey, J. Boldingh Debernard, M. Bushuk, S. Delhaye, D. Docquier, D. Feltham, F. Massonnet, S. O'Farrell, L. Ponsoni, J. M. Rodriguez, D. Schroeder, N. Swart, T. Toyoda, H. Tsujino, M. Vancoppenolle, and K. Wyser. An inter-comparison of the mass budget of the Arctic sea ice in CMIP6 models. *The Cryosphere*, 15(2):951–982, 2021. doi: 10.5194/tc-15-951-2021. URL `https://tc.copernicus.org/articles/15/951/2021/`.

J A Screen. Influence of arctic sea ice on European summer precipitation. *Environmental Research Letters*, 8(4):044015, 2013. URL `http://stacks.iop.org/1748-9326/8/i=4/a=044015`.

Petteri Uotila, Hugues Goosse, Keith Haines, Matthieu Chevallier, Antoine Barth/'elemy, Clément Bricaud, Jim Carton, Neven Fučkar, Gilles Garric, Doroteaciro Iovino, Frank Kauker, Meri Korhonen, Vidar S. Lien, Marika Marnela, François Massonnet, Davi Mignac, K. Andrew Peterson, Remon Sadikni, Li Shi, Steffen Tietsche, Takahiro Toyoda, Jiping Xie, and Zhaoru Zhang. An assessment of ten ocean reanalyses in the polar regions. *Climate Dynamics*, 52:1613–1650, 2019. doi: 10.1007/s00382-018-4242-z. URL https://doi.org/10.1007/s00382-018-4242-z.

Liping Zhang, Thomas L. Delworth, Xiaosong Yang, Fanrong Zeng, Feiyu Lu, Yushi Morioka, and Mitchell Bushuk. The relative role of the subsurface southern ocean in driving negative Antarctic sea ice extent anomalies in 2016-2021. *Commun Earth Environ*, 3:302, 2022. doi: 10.1038/s43247-022-00624-1. URL https://doi.org/10.1038/s43247-022-00624-1.

---

## Author Comment (AC2)

**Responses to Reviewer 2**
**Anatomy of Arctic and Antarctic sea ice lows in an ocean–sea ice model**

Benjamin Richaud, François Massonnet, Thierry Fichefet,
Dániel Topál, Antoine Barthélemy and David Docquier

**General Comments**

> **General Comments**
>
> The authors in this study use the NEMO-SI3 model to investigate extreme lows in sea ice extent in both hemispheres over the historical period by using sea ice mass budget terms, both hemispherically and regionally. These modeled mass budget terms are particularly valuable as they are not available in observational products but they provide insight into the processes driving the low extent years. The authors find that basal processes (melt and growth) become increasingly dominant in the mass budget in both hemispheres. Additionally, they find that during case study low extent events the importance of thermodynamics and dynamics can vary, as can the importance of difference processes by season or hemisphere due to the local ice state and drivers. Overall, this study is well designed and provides insight into how sea ice evolution is changing. I have some major concerns, detailed below, about the analysis that I would recommend being addressed in a major revision before publication.

**Response:**
We wish to thank the reviewer for their constructive remarks and believe that addressing them will improve the manuscript.
* * *
> **General Comment: Mass budget analysis**
>
> The authors present their mass budget analysis in section 4 as absolute mass loss anomalies, which itself is dependent on mean state of the sea ice. As a result, I found that the conclusions and graphs were confusing and a bit misleading about the results and more care needs to be taken with how this is presented. In years with extreme low extent anomalies the anomalously high positive ice mass budgets were sometimes explained as "preconditioning" based on the mean state. I recommend showing the anomalous mass budget terms normalized by the total ice area each year to better understand how the budget terms compare to other years. Additionally, it would be beneficial to show how the dominant budget terms change over time to support your assertions about basal melt change (e.g. Line 480-485) since Table 1 just gives a single linear trend but doesn't show the budget terms over time. It would also be helpful to show a timeseries of the fractional contributions of each mass budget term, like this figure (right) from Bonan et al. 2021 (https://doi.org/10.1088/1748-9326/abe0ec), to show how basal terms become more dominant. Again, you'd want to normalize this so that it isn't just reflecting a mean state change of less melt due to less volume.

**Response:**
We agree that positive melting anomalies are particularly difficult to interpret (see also comments from Reviewer 3). We have attempted to normalise the anomalies, either by the sea ice extent (SIE, Fig. R1b), the sea ice area (similar to sea ice extent) or the sea ice volume (SIV, Fig. R1c) in August, as it is the final state of our period of integration. Yet this raises some issues in sectors where the SIE or SIV are close to 0 at the end of the melt season, without solving the problem of positive melting anomalies in other sectors (cf. Fig. R1a). This normalization actually tends to introduce the SIE or SIV trend into the signal, exacerbating the anomalies, either positive or negative. Tests with normalization by SIE or SIV calculated at other months of the year (e.g. Apr, May, September) were not conclusive either. We therefore refrain from adding this normalization in our analysis, but will mention in the methods that we conducted inconclusive tests to normalise the fluxes.

We appreciate the suggestion to show the long-term evolution of the mass budget terms (Figure R2). We provide stacked plots for both hemispheres, for melting terms and growing terms. We normalize the terms by the total sea ice mass loss or gain calculated from a year starting in September for Arctic and February for Antarctic, to show the proportion due to each term. While we believe this is an interesting way of showing the respective proportion of the different terms and the interannual variability, we find that the trends are too small to be clearly visible on the figure, hence necessitate to be written on the graph and leading to a less legible overall presentation. Moreover, this requires a lot of panels and is not as concise and precise a way to indicate the trends as the table. We therefore lean towards keeping the table in the main text and inserting those plots in the supplementary information.

[Figure]

(a) Non-normalized     (b) Normalized by SIE (Aug)     (c) Normalized by SIV (Aug)

Figure R1: Time series evolution of mass budget terms in Arctic during the melt season (May to August): a) non-normalized terms (as used in the original manuscript), b) normalized by the minimum (August) sea ice extent (SIE) and c) normalized by the minimum (August) sea ice volume (SIV). In sectors and years when the SIE or SIV are close to 0, the normalization leads to very large values, preventing any comparison with other years. In other sectors, the normalization introduces a trend in the signal, due to the decreasing SIE and SIV over the total period.
* * *
**General Comment: Ocean Heat Content**

OHC is mentioned several times (Line 353, 358, 455, 454, 466) but is not clearly shown in figures and needs more analysis to support your assertions. For the Arctic cyclone in 2012, you should have a figure like Fig. 6 that shows the OHC aspect of this story and the impact on basal melt. For the Antarctic, you should do a spatial correlation of OHC and ice anomalies to quantify the relationship described at Line 445. Since you're analyzing winter months, there may be important implication of OHC anomalies that are not just from surface heating or the albedo effect. Additionally, for the panels in Figure 6, you say you are integrating OHC south of 50S. In many of the sectors this means you're including anomalies that are not actually in contact with the sea ice since they're well north of the seasonal sea ice zone. This makes your analysis at Lines 454-456 and 466-467 of the anomalously low OHC in the Ross-Amundsen sectors confusing since much of the anomalously low areas are not in contact with sea ice. I recommend you mask Figure 6 to show only that year's sea ice zone. Additionally, you may want to calculate a regional mean OHC anomaly only over points that are within the SIZ to better understand how the sea ice will be affected by OHC year to year.
* * *
**Response:**

We appreciate the reviewer's comment, which is also supported by Reviewer 3's suggestion. We believe that properly addressing both reviewers' comments would require to delve into a long investigation regarding causality between OHC and basal melt. Another in-depth study using advanced statistical methods, namely the Liang-Kleeman information flow,

(a) Arctic

(b) Antarctic

Figure R2: Long-term evolution of mass budget terms, expressed as a proportion of total sea ice mass gain for growing terms (blue shades, left columns) and of total sea ice mass loss for melting terms (red shades, right columns), for the Arctic (a) and the Antarctic (b). Percentages indicate the long-term trend for the different terms, in bold font when significant (p-value < 0.05), expressed in % per decade (as in Table 1 of the original manuscript). Melting residuals are not plotted as they represent less than 1 % of total mass loss and are therefore not visible anyway. Trends for growing residual (porous ridging) are not indicated to avoid overcharging the figures.

is precisely in preparation, led by one of the co-authors, to convincingly disentangle the causality between OHC and basal melt. Diving into this would therefore be redundant with the study in preparation, would take too much space and effort relative to the rest of this manuscript, and would also be out of the scope of this study which focuses on the sea ice mass balance.

We therefore suggest to simply remove most of the analysis around the ocean heat content, including Fig. 6 and the paragraphs l.436-476. We will remove mentions of OHC when mentioning the lack of impact of the 2012 storm (l.349-359) and will shorten the part in the Discussion section (l. 551-561) to a couple of sentences simply mentioning the fact that the model reproduces the documented OHC increase in both the Beaufort Gyre and the Southern Ocean, and will simply refer the reader to the literature analysing the link between subsurface ocean heat content and transport and the sea ice lows.

We hope this addresses the reviewer's concern.

> **General Comment: Antarctic transport**
>
> Antarctic transport: The discussion from Lines 229-241 on Antarctic transport was confusing. You mention the "circumpolar current", which probably refers to the ACC that is on the northern edge of the sea ice zone. What about transport from other currents like coastal currents which are in the reverse direction? I think that the transport piece itself is complex and needs to be more carefully assessed and discussed.

**Response:**

We agree that the transport is difficult to analyse properly. We will correct this analysis by clarifying that we indeed talk about the ACC, and by also mentioning the westward coastal currents. Their effects on ice mass budget would be similar, because ice inflow coming from an eastern sector would be compensated by an ice outflow towards a western sector, except in regions of re-circulation (e.g. Weddell Sea). We will discuss this aspect in the manuscript. We fully agree that the transport piece is complex, and we did our best to address it properly when it becomes one of the dominant processes.

> **General Comment: Clarity**
>
> Clarity: Much of the paper consists of VERY long paragraphs that are difficult to follow for a reader and often contain several points (e.g. Lines 132-162, 368-393, 402-435, section 5). Please break these paragraphs up so that they're clearer and more honed to the points you are making. A few places I found particularly challenging to follow were Section 4 and the discussion conclusions. Section 4 is framed as case studies of different hemispheres and years. I think it could be more powerful to frame these in terms of processes. For example, for section 4.1 a title like "the impact of ice volume on extremes" will help underscore the conclusion is that sea ice extent alone does not tell the full story and volume is necessary. Section 4.2 seems to be comparing hemispheres, but depends on how free drift and dynamics vs. thermodynamics impacts the extremes. Section 4.3 is better in that it clarifies seasonal differences are the focus. You may want to compare seasons for the Arctic too to confirm your results and see if the processes driving each hemisphere differ similarly by season. The discussion and conclusions sections are wordy with long paragraphs that make it difficult to separate the primary conclusions from the discussion. Perhaps make conclusions its own, small final section and better hone the paragraphs in this entire section.

**Response:**

We agree with the reviewer that many of our paragraphs are too long. We will do our best to split the paragraphs into shorter paragraphs and to clarify our writing.

Following the reviewer's suggestion, we will modify Section 4 by framing it in terms of processes, modifying the titles to match the main results of each subsection and reformulating the conclusions of each subsection. We might keep the mention of the comparisons as subtitles of each subsection, as we believe this could also help the reader to anticipate the content of the subsection. We refrain from adding a comparison to an Arctic winter, as those winter conditions are strongly constrained by geography; winter 2006 could be an interesting case study but the ice loss is limited to the Labrador and Greenland Seas, which would require to include a discussion on deep convection and OHC, which we excluded following the reviewer's suggestion. Winter 2025 would be great to analyse, but our model outputs do not (yet) extend to 2025.

We will also implement the reviewer's suggestion to split Section 5 into two sections by

merging the second and third paragraphs into a Discussion section and the first and fourth paragraphs into a Conclusion section. We will split the Discussion section into two subsections, the first one addressing discussion of the processes (equivalent to the current third paragraph of Section 5) and the second addressing limitations of the model and methods (equivalent to the current second paragraph of Section 5).

**Specific Comments**

> **Comment 1**
>
> Figure 1: Since a big part of this paper is showing that sea ice volume or mass is critical for understanding extreme ice extent loss events (e.g. line 166, 322, 326, etc.), you should include panels with the timeseries of volume anomalies in this figure. These are already Figures S1a and S4a, but they should be included as main figure panels as well. Additionally, it could be beneficial to compare to a hindcast that uses data assimilation (e.g. PIOMAS for the Arctic, SOSE for the Antarctic?) to see how well your model compares to some estimates of these variables.

**Response:**
We will add two panels to Fig. 1 (see updated version Fig. R3) showing the sea ice volume anomalies from the model and from PIOMAS and GIOMAS (SOSE does not seem to extend prior to 2008). We will comment those two panels in the main text in parallel of the comparison of the sea ice extent (l. 146-159).

> **Comment 2**
>
> Figure 1: Panels a and c are averages over some years – you should list the years they are averaged over in the figure caption and show the standard deviation to indicate variability.

**Response:**
We will make these modifications (see Fig. R3 for standard deviation).

> **Comment 2**
>
> Figure 3: This is a great figure! The map legend doesn't show white on white for the contour, so this should be fixed.

**Response:**
Thank you. This mistake will be corrected by adding a gray background to the legend (Figure R4).

> **Comment 2**
>
> Figures 3/4/5/6: These figures are nice, but the graphs overlain on the maps makes them hard to read, especially the center Antarctic one. Additionally, there are no y axis labels on the regional graphs. For figures 4/5/6, the bar charts axes limits aren't consistent, so it is hard to compare the panels with one another. Consider have a white, opaque background or reorganize the figures so that the graphs are separated from the maps and easier to compare, and please be sure to label axes.

**Response:**
We will improve those figures by adding a white background behind each inset panel (Figs. R4, R5 & R6). We refrain from adding a y-axis label to all regional panels, as it is the same for all and would therefore be redundant while cluttering the figures. We will clarify that the y-axis label and units are the same for all panels in the figure caption.

**Figures: new versions**

[Figure]

Figure R3: Figure 1, new version. Modified caption:
Comparison of observations-based products (black) versus model sea ice extent (red) and volume (blue). Observations-based products are the satellite-based NOAA/NSIDC Climate Data Record (CDR) for sea ice extent and the PIOMAS and GIOMAS reanalyses products for sea ice volume. Mean seasonal cycle of sea ice extent for (a) Arctic and (d) Antarctic. Shading indicates one standard deviation. Minimum sea ice extent anomalies relative to the mean seasonal cycle for (b) September in Arctic and (e) February in Antarctic. Minimum sea ice volume anomalies relative to the mean seasonal cycle for (c) September in Arctic and (f) February in Antarctic; correlations between observations and model are given in the top-right corner of each panel for the minima comparisons (panels b, c, e and f). Note that the time values include the month, meaning that the point for e.g. September 2000 is closer to the x-tick value 2001 than 2000. For more robust comparisons, all mean seasonal cycles used in this figure are calculated over the whole available satellite period (1979-2023).

[Figure]

Figure R4: Figure 3, new version.

[Figure]

Figure R5: Figure 4, new version.

[Figure]

Figure R6: Figure 5, new version.

---

## Author Comment (AC3)

**Responses to Reviewer 3**
**Anatomy of Arctic and Antarctic sea ice lows in an ocean–sea ice model**

Benjamin Richaud, François Massonnet, Thierry Fichefet,
Dániel Topál, Antoine Barthélemy and David Docquier

**General Comments**

> **Manuscript Synopsis**
>
> This manuscript, using an atmosphere forced ocean-ice historical reconstruction attempts to analyse sea ice flux mass contributions for exceptional sea ice events relative to climatology. The paper uses the common methodology to compare and contrast sea ice events in the Arctic and Antarctic, as well as comparing/contrasting melt and freeze-up season events. The paper could provide a useful measure examining exceptional sea ice events, but suffers considerably by confusing and incomplete graphics as outlined in my major comments below. In particular, the pre-existing sea ice anomaly plays a huge role in allotment of mass balance fluxes into anomalous fluxes, particularly for melt events, where there is a definitive upper bound on sea ice removal (you cannot remove more ice than what already exists). The authors discuss this "pre-conditioning" (their term) in the text, but the lack of a graphical representation of this term in their budget can lead to confusing interpretation of the results, particularly by a reader seeking quick visual summarization of the results.

**Response:**
We thank the reviewer for their thoughtful and thorough revision. We address their comments below and believe implementing them will significantly improve the manuscript.

> **Major Comment 1**
>
> Fig. 4 & 5. I found the presentation of these figures very hard to follow, as the discussion relies heavily on an additional term (preconditioning or initial mass anomaly) that requires careful reading of the manuscript to draw out. As the figures stand now, it is very easy to convince oneself that positive flux anomalies mean a decrease in sea ice mass, when in actual fact they mean the opposite (increase in sea ice mass), but in a vast proportion of the sea ice thickness anomalies, this incorrect assumption does seem to visually confirm – and in some cases (Labrador Sea / Baffin Bay) seems to be wholly nonsensical. To make graphical interpretation much simpler:

**Response:**
We thank the reviewer for their comment. This major comment is at odds with comments from the other reviewers, which makes it particularly difficult to address. Please note that the other reviewers praised those two figures and don't seem confused by the sign

convention. We have attempted to find a middle ground. We provide more details below when addressing the detailed comments from the reviewer.

> The sea ice thickness anomaly must be over the same period (May to August) as the flux anomalies (and not just August).

**Response:**
We do not think implementing this suggestion would be appropriate. Showing the mean anomaly over the whole melt season would prevent any direct comparison between the map and the mass flux anomalies in the insets. Indeed, we are here calculating a mass budget, integrated over the melt season (and expressed as anomalies). Following this method, the ice mass budget explains the *difference* of ice mass between the beginning and the end of the melt season, not the *mean* state during the melt season:

$$\int_{May}^{Aug} \frac{\partial M}{\partial t} = M_{Aug} - M_{May} = \int_{May}^{Aug} \Sigma F_{mass} \tag{1}$$

In the current background map of Fig. 4, we plot a proxy (the thickness instead of the mass, as is standard in the literature) of the anomaly of $M_{Aug}$ while the anomaly of $M_{May}$ corresponds to what we call the preconditioning. We could potentially replace the thickness anomaly by the difference in state between May and August, but consider this would make the plot less easily comparable to other studies or data sources while providing a limited improvement in the interpretation of the results. As suggested below by the reviewer, we instead provide the preconditioning by sector in the inset panels.

> This still does not lead to a visual flux closure as it does not account for the initial sea ice mass anomaly (pre-conditioning in the author's terminology), which in many, if not all cases is the main offsetting factor. Therefore an additional "preconditioning" pseudo-flux should be added to the bar chart representing the initial mass anomaly (technically it should be for 30 April, but an average over April should be close enough if more convenient/smoother). I initially though you would need to convert this into a flux – but if I understand correctly the fluxes are already time integrated into mass gain over the 4 months? [You would likely not wish to add this pseudo-flux to the total, just leave it separate.]

**Response:**
We appreciate this suggestion and will include the mass anomaly on the first day of the period of interest (May 1st for Arctic, October 1st for Antarctic Summer 2022 and April 1st for Antarctic Winter 2023) as a dashed black line in the insets of Fig. 4 & 5, instead of in the Supplementary information (see Figs. R3 & R3). We will also clarify in the figure caption that "the net mass anomaly for each sector at the end of the period of interest can be estimated by summing the preconditioning anomaly (dashed line) and the total mass flux (black bar)." We hope this will improve the interpretation of the figure. We believe it now makes our initial analysis clearer, by showing that the total positive flux anomaly in sub-Arctic seas is of similar amplitude as the negative preconditioning term, except in the Greenland Sea sector in 2012 where advection plays a important role, as already properly mentioned in

the initial manuscript. We will correct our analysis of the Beaufort Sea in 2007, as this new format seems to indicate that preconditioning could play a role, though by changing the dynamical term rather than the thermodynamical ones. We also see a good agreement for Weddell Sea in 2022, validating our initial assessment.

> Only then will it the figure visually balance the fluxes in the sector with the mass loss/gain contours.

**Response:**
We agree that adding the preconditioning now leads to a closure of the budget.

> It will also visually confirm large segments of the text which discuss that the apparent flux anomaly is actually due to "pre-conditioning" (i.e. the initial mass anomaly), with the increase in anomalous ice mass fluxes (i.e. mass growth) being largely offset by the initial anomaly. In other words, there is an increase in anomalous ice growth largely due to there being less ice than climatology to melt!

**Response:**
This was indeed our initial interpretation, and we believe the reviewer's suggestion to add preconditioning in the insets now makes it clear.

> Ultimately, the usage of anomalous fluxes seems to be less than informative, the size of the flux ultimately being hugely dependent on the underlying sea ice volume. A better strategy (with no guarantee of success) might be to use normalized (either by total ice volume, or ice volume change, the latter assuming a definitive melt/freeze sign by sector) fractional flux anomalies. For instance does the fraction of basal sea ice melt increase or decrease from climatology in the exceptional years? Note: The fractional flux could be greater than 1, or less than zero. Sign conventions, for lack of better terminology, might be messy. I do not suggest pivoting to such an analysis now, I would view this manuscript as a learning process in best practices in this regard.

**Response:**
This is also a suggestion from Reviewer 2, that we have explored (see response to Reviewer 2, Figs. R1 & R2). The normalization introduces significant challenges that make the interpretation of the results even more confusing and complicated than the current methodology, to our opinion. Expressing the mass fluxes as a proportion of the total annual ice loss or growth (Fig R2) is a reasonable approach to investigate interannual variability. Yet, a major caveat of this approach is that the transport term cannot be included, as it can be either positive or negative and therefore cannot be expressed as a percentage of total mass loss or gain.

We like the reviewer's view that this manuscript can participate to the learning process of how to best investigate budget analyses. It is worth noting that this is not specific to ice mass budgets, and is a shared issue with other topics, such as ocean heat budget analyses (e.g. for marine heatwaves).

Examples of confusing aspects:

- (a) Erroneous statement of Major Comment 3.

- (b) Statement concluding Subsection 4.1 (Minor Comment 1)

- (c) Using text explanations to highlight effect of pre-existing mass anomaly (preconditioning) without additional graphical assistance (ll. 311, 314, 338, 360, 363, 383, 394, 489–505).

- (d) Large sea ice growth flux anomaly in Baffin and Hudson Bays/Labrador Sea sector with only a small manifestation of sea ice loss in the Canadian Archipelago.

**Response:**
See Major Comment 3 and Minor Comment 1 for our response. For the text explanations of preconditioning (point c), we will now be able to refer to Fig. 4 & 5 explicitly and hope this makes the text clearer. We do not understand the reviewer's point d, as there is no growth anomaly. Did the reviewer mean a positive melt anomaly? If so, first, the absolute value of those anomalies is relatively small compared to other sectors, and this means a lack of melt that is clearly explained by the pre-conditioning term, as initially suggested.
* * *
**Major Comment 2**

Ocean Heat Content: I am not entirely convinced of all the claims made in the manuscript with regards to increased heat content leading to increased basal melt. (a) The stated alignment of the increased heat content (Figure 6; red; numerically positive) and decreased sea ice volume (Figure 5b; red, numerically negative) do not line up as well as suggested as demonstrated in the enclosed animated gif which purposes to overlay the two (I see a lot of alternating red/blue). Caveat: As with my comments with regards to Figures 4 & 5, the heat content (Figure 6; April to September) does not align in time perfectly with the sea ice volume (Figure 5b; September) either. (b) The choice of the 100-200m heat content is a little confusing, and not justified. The winter time mixed layer depths [Uotila et al., 2019] range from 100 to 300m, which means an increase or decrease in the mixed layer could have opposing tendencies in the top 100m and 100-200m – if the mixed layer increases one might expect the upper layer to warm while the lower layer cools (increased surface mixing with the warmer below mixed layer waters), with the opposite cooler surface, warmer 100-200m if the mixed layer decreases (isolates the surface). (c) The previous point is very well illustrated in Figure 2 of the reference Zhang et al. [2022]. Indeed, the zero lag in that paper would seem to require an accompanying negative anomaly in 100-200m heat content. The mechanism also requires a long period lagged relationship that I see no evidence of here. (d) I might speculate that the heat contents are sea ice driven: Lower sea ice creation implies lower brine rejection and increased stratification (isolation) of the surface waters, increasing the 100-200m heat content. This would explain the relative uptick in Antarctic September heat content relative to April heat content seen at the end of the pan-Antarctic time series in Figure 6 – but it is also difficult to see if this is an isolated event, or a common occurrence. (e) As the authors state, the increased/decreased basal melt/growth may be driven by the atmospheric forcing, especially in low sea ice thickness states as the downward heat fluxes directly heat the ocean surface layers. (f) I do not advocate that my speculations, or any of the alternative explanations are more or less likely than the mechanisms suggested by the authors. I do suggest there is a lack of current evidence in the manuscript for any conclusions connecting the heat content to the loss of sea ice volume. (Seasonal) Lead/lag relationships may be critical.
* * *
**Response:**

We appreciate the reviewer's comment, which is also supported by Reviewer 2's comments. We agree with the overall statement that we do not provide a strong proof that OHT leads to sea ice mass decrease, though we tried to avoid making such a claim in the initial manuscript and attempted to rather suggest that out model is in line with other studies. In any case, we believe that properly addressing both reviewers' comments would require to delve into a long investigation regarding causality between OHC and basal melt. Another in-depth study using advanced statistical methods, namely the Liang-Kleeman information flow, is precisely in preparation to convincingly disentangle the causality between OHC and basal melt. Diving into this would therefore be redundant with the study in preparation, would take too much space and effort relative to the rest of this manuscript, and would also be out of the scope of this study which focuses on the sea ice mass balance.

We therefore suggest to simply remove most of the analysis around the ocean heat content, including Fig. 6 and the paragraphs l. 436-476. We will remove mentions of OHC when mentioning the lack of impact of the 2012 storm (l. 349-359) and will shorten the part

in the Discussion section (l. 551-561) to a couple of sentences simply mentioning the fact that the model reproduces the documented OHC increase in both the Beaufort Gyre and the Southern Ocean, and will simply refer the reader to the literature analysing the link between subsurface ocean heat content and transport and the sea ice lows.

We hope this addresses the reviewer's concern.
* * *
**Major Comment 3**

Erroneous statement: l. 328. The manuscript states there is an increase in basal melt in the Chukchi Sea in July 2007. Supplementary figure S2m-o shows a blue (positive) basal melt anomaly in the Chukchi Sea. But positive flux anomalies are defined as anomalous gain of ice mass. Therefore this is not an increase in basal melt, but a decrease in basal melt. If I am incorrect, please correct me, but this does demonstrate my confusion generated by the figures. I suspect this positive basal melt anomaly is completely due to "preconditioning," – i.e. there is a anomalous lack of sea ice to melt.

- Similarly, the Chukchi and Bering Seas sector shows a net positive basal melt flux (so again decreased basal melt) in the bar charts of Figure 4a.

- ll. 329. If I am not confused, and the basal sea ice melt is actually decreased, the connection to ocean heat transport may no longer be appropriate, however, you should have stated (the perhaps obvious, nevertheless still useful) that there are observations of increased northward heat transport. I briefly contemplated the authors meant there was an observed southward transport of heat to match the flux anomaly.

- The statement "this increase (in sea ice mass) is not sufficient to compensate the export of ice" is correct – but again only added to my level of confusion.

- Please check your characterization of your flux sign convention elsewhere in the manuscript. Having noticed this, I cannot convince myself there may be other instances where I have matched my interpretation of the sign convention in the graphics to match the text commentary (i.e. I can be easily confused into agreement).
* * *
**Response:**

We believe there was here a lack of clarity from our part on the exact location of the anomaly, rather than an erroneous interpretation. We were referring to the small red anomaly visible in July (and August, though less clear) just offshore/north of the Chukchi Sea, over Barrow Canyon. Barrow Canyon is a known hotspot for advection of Pacific water into the Central Arctic. We interpret this negative basal melt anomaly (also slightly visible in the surface melt as a lack of anomaly) as increased warm inflow, matching the observation-based documented inflow (Woodgate et al. 2010). This was a clear mistake from us to not mention the exact location of this negative anomaly, and we will correct it in the revision. We hope this answers Major Comment 3.

**Minor Comments**

**Comment 1**

ll 362-363: "It (2012) is therefore a low not only in sea ice extent, but also in volume, in contrast with summer 2007." This comment cannot be made here without specifying you are excluding the seasonally ice covered Labrador Sea / Baffin Bay and Greenland Sea and Barents-Kara Seas sectors as previously mentioned in the text. Readers just reading the section concluding remarks (it does happen) will immediately refer to figure 4 and both conclude you have this backward – 2012 has no change in volume, and 2007 has a low in sea ice volume (i.e. invert your sign convention). But if you also include the pre-conditioning flux this will also be rendered visually correct.

**Response:**

While we might not fully understand the reviewer's comment, we do not think that excluding the sub-Arctic Seas (Labrador Sea/Baffin bay, Greenland Sea and Barents-Kara Seas) is necessary for our statement to be true. First, Fig. S1.a, which shows the SIV including the subarctic seas, indicates a minimum in September 2012, justifying our statement. Second, those seas are almost ice free in a climatological sense at the end of the melt season. Therefore, including or excluding them does not change significantly the estimate of the August sea ice volume nor extent. We think that including the preconditioning in Fig. 4 clarify this, as anticipated by the reviewer.

**Comment 2**

ll. 81-82. There are considerably more examples and research concerning climatic implications of changes to sea ice [e.g. Screen, 2013] – I would normally provide a more extensive list, but I am stressed for time here (no conflicts in solitary suggestion).

**Response:**

We thank the reviewer for the suggested reference that we will incorporate, along with others (including Honda et al. 2009, Strey et al. 2010). Note that many of the references investigating the climatic impacts of sea ice loss tend to focus on long-term (multi-year) loss, rather than a single-year minimum.

**Comment 3**

ll. 110-111. The tri-polar grid is designed to remain eddy-permitting in the Arctic (grid cells of order 12km). I should probably know this, but even so, others readers might not. Does the eORCA025 grid remain eddy-permitting throughout the Antarctic domain?

**Response:**

Yes, it remains eddy-permitting in the Antarctic domain, with an effective resolution around 7 km in the Ross and Weddell Seas, which is slightly lower than the Rossby deformation radius in those regions.
* * *
**Comment 4**

l. 117. Is it standard to have equal numbers of sea ice and snow layers (2+2)? I obviously do not know, but I seem to recall the multi-layer thermodynamics sea ice models I have dealt with have more sea ice layers than snow layers. Is there a rationale for this?

**Response:**

This is the default setup in NEMOv4.2.2 and we kept it that way. Other configurations with more sea ice layers than snow layers exist (e.g 2+1; 10+5 as the new default in NEMOv5), but we do not know of any definitive rule about this.
* * *
**Comment 5**

l. 151: entire time series. I assume this is your entire analysis time period (1979-2023), but perhaps it is worth repeating here? And the last two decades are presumably 2004-2023?

**Response:**

We will clarify that: "[...] when calculated over 1979-2023, but increases in the Antarctic when considering only 2004-2023."
* * *
**Comment 6**

Figure 1b/d: From the looks of the plot, I assume the numerical minimum for both the Arctic and the Antarctic has a time value assigned by exact time in year (year + month + day), or in other words, the Arctic sea ice minimum for 2009 is more closely aligned (2009.75, or slightly to the left) of the 2010 grid line than the 2010 (2010.75) minimum. The same applies to the Antarctic, but in this case it is more closely aligned with the correct calendar year. If so (or if not) this should be communicated in the caption. The same question can apply to Figure 6 – with not as much consequence – are the annual mean (+0.5) aligned in time with April (+0.25) and September (+0.75), or are they offset by 0.25? Similarly figures S1 and S4.

**Response:**

This is indeed the case, meaning the September year X anomaly is indeed closer to year X+1. We will clarify this in the caption: "Note that the time values include the month, meaning that the point for e.g. September 2000 is closer to the tick value 2001 than 2000."
* * *
**Comment 7**

l. 171. Units are slightly confusing, suggest reordering somewhat (in kg for sector analysis, and kg m-2 for individual grid points). However, it is also important this information be added to the figure captions – at least in the first instance of occurrence in a figure. (Figure 3, 4, 5 for sector flux values; Already included for grid values (S2, S3, S5, S6).

**Response:**

We will implement this suggestion. Note that in the figures, we actually converted the kg to Gt to avoid exponents and to make it easier to compare to other studies.
* * *
**Comment 8**

l. 192. Keen et al. [2021] is an extension of Keen and Blockley [2018] to a multi-model analysis. I would think that (no conflicting interest) the original budget analysis would be a more appropriate citation. Keen et al. [2021] would remain applicable for placing this manuscript's results within the CMIP context (ll. 224, 226, 289, 290). Note the DOI is correct (https://doi.org/10.5194/tc-15-951-2021, but the link does not work properly (across two lines, with the automatic line numbering interfering?) for Keen et al. [2021].

**Response:**

This is a good suggestion, we will incorporate it. We expect the broken doi to be fixed at the proofreading stage.
* * *
**Comment 9**

The sector Labrador Sea and Baffin Bay also includes Hudson Bay, which is likely an equal contributor to the sea ice mass changes over the May-August period. While Labrador Sea / Baffin and Hudson Bays is likely too lengthy for labelling purposes, I favour the more accurate East Canada Arctic moniker (Eastern Canadian Arctic is more grammatically correct, but longer).

**Response:**

We agree with the reviewer and will rename this sector to "East Canadian Arctic". The name "Labrador Sea-Baffin Bay" is the one used in Koenigk et al. (2016) from which we took the region definitions: we will clarify in the text that we have modified the nomenclature for accuracy purposes.

**References**

Honda, M., Inoue, J., and Yamane, S.: Influence of low Arctic sea-ice minima on anomalously cold Eurasian winters, Geophysical Research Letters, 36, `https://doi.org/10.1029/2008GL037079`, 2009.

Strey, S. T., Chapman, W. L., and Walsh, J. E.: The 2007 sea ice minimum: Impacts on the Northern Hemisphere atmosphere in late autumn and early winter, Journal of Geophysical Research: Atmospheres, 115, `https://doi.org/10.1029/2009JD013294`, 2010.

**Figures: new versions**

[Figure]

Figure R1: Figure 1, new version. Modified caption:
Comparison of observations-based products (black) versus model sea ice extent (red) and volume (blue). Observations-based products are the satellite-based NOAA/NSIDC Climate Data Record (CDR) for sea ice extent and the PIOMAS and GIOMAS reanalyses products for sea ice volume. Mean seasonal cycle of sea ice extent for (a) Arctic and (d) Antarctic. Shading indicates one standard deviation. Minimum sea ice extent anomalies relative to the mean seasonal cycle for (b) September in Arctic and (e) February in Antarctic. Minimum sea ice volume anomalies relative to the mean seasonal cycle for (c) September in Arctic and (f) February in Antarctic; correlations between observations and model are given in the top-right corner of each panel for the minima comparisons (panels b, c, e and f). Note that the time values include the month, meaning that the point for e.g. September 2000 is closer to the x-tick value 2001 than 2000. For more robust comparisons, all mean seasonal cycles used in this figure are calculated over the whole available satellite period (1979-2023).

[Figure]

Figure R2: Figure 3, new version.

[Figure]

Figure R3: Figure 4, new version.

[Figure]

Figure R4: Figure 5, new version.

---

## Author Comment (AC4)

**Responses to Reviewer 1**
**Anatomy of Arctic and Antarctic sea ice lows in an ocean–sea ice model**

Benjamin Richaud, François Massonnet, Thierry Fichefet,
Dániel Topál, Antoine Barthélemy and David Docquier

**General Comments**

> **General Comments**
>
> Richaud et al. present an analysis of record lows in Arctic and Antarctic sea ice extent using mass balance decomposition in an atmosphere-forced ocean–sea ice simulation. Their findings highlight that interactions at the ice–ocean interface are common to the four case studies examined, and that the role of basal melt in particular is increasing with climate change. The main novelty of this study is that different sea ice low events from both hemispheres are analysed in a self-consistent framework, which is distinguished from previous literature examining single cases using a variety of observational data sources and/or model configurations. I agree this approach has merit. While there is not much new insight into the individual cases (which have been studied extensively), the more generalised understanding and comparison of sea ice lows makes up for this. The manuscript is structured logically overall, and the figures are very clear. I like the design of Figs. 4 and 5 in particular, reflecting nicely the use of "anatomy" in the manuscript title.

**Response:**
We thank the Reviewer for their thoughtful review and believe that their comments and suggestions will help us improve the manuscript.

I would appreciate some clarification on how anomalies in the mass budget terms are calculated. This may reflect a misunderstanding on my part or a potential issue in the methodology. Lines 196–204 explain how the climatological seasonal cycle is calculated, but it does not seem like the long-term trend is removed from the data. This would cast doubt on the magnitude and sign of the anomalies presented in Fig. 4 (and, to a lesser extent, Fig. 5), and hence the interpretation, depending on the strength of the trends in each term. For example, Fig. 1.b has clearly not been detrended. In that case it doesn't really matter for the purpose of the plot, but if the corresponding time series for the mass budget terms looked like this then everything after roughly the mid-point of the time series would be a negative anomaly. Despite all this, Fig. 4 clearly show anomalies of both sign, and the resulting interpretation is consistent with previous studies, which suggests either (1) the trends are sufficiently small for this to not matter too much or (2) the trends have actually been removed. I would have thought the trends should be removed (and, if only stylistically, also for Fig. 1b, for consistency with the description of a sea ice low on l.38 as "when the sea ice extent becomes significantly lower than the trend line").

**Response:**

As the reviewer supposed, we do not remove the trend for the mass budget terms, for two main reasons.

1. The first one is physical and is a question of perspective: one can consider that what really matters is the specific short-term event(s) of the year, therefore excluding the long-term trend from the anomalies, as suggested by the reviewer. But one can also consider that the external forcing represented by the trend is part of the causes of sea ice lows, and that it should therefore be included in the anomalies (as we do here). Both perspectives are valuables, have pros and cons, and the choice between both methods should be guided by the overarching research question. In our case, we compare both hemispheres and different seasons, on top of different years, and consider that the differing trajectories in sea ice extent between the Arctic and the Antarctic are an interesting component of the causes of the sea ice lows. We therefore prefer to keep the trend in the signal. Note that this is reminiscent of the considerations around the baseline choice in the marine heatwave literature (Smith et al. 2025).

2. The second reason for keeping the trends in the signal is a technical reason. The trends are highly seasonal: for example, the melting terms are null during the whole growth season, meaning there is no trend for that time of the year, while there can be some trend during the melt season. Removing an annually calculated trend (linear or high order) would then lead to non-null values for the melting terms during winter and for growing terms during summer (Figure R1). This is unphysical and more difficult to interpret and justify. Removing a seasonally-varying trend would only partially solve the problem, as it would still lead to an arguable, positive value for the melt term at the growth-melt transition period if melting starts later than usual. This would add significant complexity not only to the method, but also to the interpretation of the results and of the trends, as each term would require its own seasonality.

For those two reasons, we refrain from removing the trend from the anomalies. We acknowledge that the manuscript did not clearly state this choice and we will correct this

omission by adding the following sentence in the methods:

l. 201: "The trends are not removed from the anomalies, to compare the impact of the differing trajectories of sea ice extent between Arctic and Antarctic. This means that the long-term changes are included in the analysis of sea ice lows."

[Figure]

Figure R1: Examples of linearly detrended anomalies, for a) basal melt in the Central Arctic and b) basal growth for the Total Arctic. A spurious positive trend seems to emerge, but is actually due to the artificial creation of a positive trend only during the winter for basal melt and summer for basal growth, seasons during which the anomalies are supposed to be null. Inset panels provide a zoom over shorter periods (2022-2024 for panel a, Apr to Sep 2007 for panel b) to exemplify the impact of removing the trend. The blue lines in the inset panels show the non-detrended anomalies (as used in the manuscript).

> Also, while the writing is clear enough, there is a somewhat common thread of vague statements that should be clarified, expanded upon, or removed (see specific comments). If the authors are able to address these among other (overall minor) concerns, I would be happy to see the study published in The Cryosphere.

**Response:**
We will simplify the writing and clarify any vague statements.

**Specific Comments**

> **Comment 1**
>
> L10: "The Antarctic 2022 event was partly driven by a strong interplay between dynamic and thermodynamic processes": this is vague.

**Response:**
We will clarify the sentence as follows: "The Antarctic 2022 event was generally driven by dynamic processes transporting sea ice towards sectors where more melt occurred."

> **Comment 2**
>
> L13–14: "highlights the potential of the ice mass budget decomposition [...]": do you mean in terms of applying such a decomposition to observations specifically (linking to the final paragraph of the discussion)? Not clear what this means otherwise, as mass budget decomposition is a fairly standard technique.

**Response:**
We agree with the reviewer that the ice mass budget has become a fairly standard technique, but not systematically used nonetheless, and often simplified as a dynamic versus thermodynamic contribution, while the decomposition into basal versus surface terms is less often used. We will clarify this as follows:
l.14 "highlights the potential of the ice mass budget decomposition to disentangle oceanic and atmospheric contributions in the evolution of the sea ice state in a changing climate."

> **Comment 3**
>
> L37–39: this is not a definition: what is "significantly lower" or "noticeable"? I suggest removing "defined here as", e.g.: "[...] occurrence of sea ice lows—instances when the sea ice extent becomes [...]". Then the next sentence then follows naturally as-is, without having to give a specific definition of "sea ice low" at all.

**Response:**
We will implement this suggestion.

> **Comment 4**
>
> L44: "was quickly ruled out (Schweiger et al., 2008)": I think it is important to note that this is according to a modelling study.

**Response:**
We will mention this, l. 43-44: "[...] was quickly ruled out by a modeling study (Schweiger et al. 2008)"

> **Comment 5**
>
> L50–51: again, I think it is important to note that this is according to modelling studies (as in both references here).

**Response:**
We will specify the data sources for those studies:

l. 49-51: "Studies based on reanalysis data suggested that a strong summer cyclone occurring in August could have played a role (Simmonds and Rudeva, 2012; Lukovich et al., 2021), but modeling investigations argued that the impact of this cyclone on ice loss was minimal (Zhang et al., 2013; Guemas et al., 2013)."
* * *
**Comment 6**

L53–54: I suggest adding the month in brackets, as done earlier in the sentence for spring.
* * *
**Response:**
We will add the month, l. 54: "in austral summer 2017 (February)"
* * *
**Comment 7**

L74–77: (i) I'm not sure it can be both "striking" and "expected"; I would just go with "striking". (ii) I would suggest rephrasing the first part to emphasise that both cyclonic and the opposite anticyclonic conditions are plausible drivers. (iii) Remove "etc.".
* * *
**Response:**
(i) We will remove "though expected". (ii) We will rephrase, l.73: "the diversity of candidate drivers is striking: both anticyclonic (in 2007) or cyclonic (in 2012) atmospheric conditions, [...]". (iii) We will remove it.
* * *
**Comment 8**

L71: Suggest starting a new paragraph with "While come causes [. . . ]", as you are now discussing both Arctic and Antarctic sea ice low events whereas the current paragraph is just discussing the latter.
* * *
**Response:**
We will split this long paragraph as suggested.
* * *
**Comment 9**

L100: "It also highlights [. . . ]": this sentence summarises findings, whereas the previous sentence explains methodology. Suggest rephrasing this to, e.g., "Our analysis reveals [. . . ]".
* * *
**Response:**
We agree, l.100: "Consequently, our analysis highlights the dominant role [...]"
* * *
**Comment 10**

L106: "Section 5 [. . . ] provides a perspective [. . . ] on the potential for future sea ice lows": there is no such perspective given in section 5. The only part of section 5 relevant to this is the reiteration of the trends in mass budget terms (L481–484), but there is no discussion on the potential of future sea ice lows. In my view, this is a key aspect missing from the discussion section and should be added (e.g., a paragraph, perhaps beginning with the information from L481–484, and linking in the results of section 3).
* * *
**Response:**
This was indeed an involuntary omission that we will correct by adding the following

sentences in the discussion:

"[...] in order to reach a state that is statistically less likely. On this basis, a number of criteria can help us anticipate future sea ice lows. First, the thinning and shrinking of sea ice leads to more potential for preconditioning through thinner ice that can melt away earlier in the season. Thinner ice is also more mobile because of a lower tensile strength, potentially increasing its dynamics (Olason & Notz, 2014; Docquier et al., 2017). Heat conductivity is higher through thin ice and therefore modifies the thermodynamics and the melt onset (Bitz & Roe, 2004). Considering that all investigated sea ice lows are a combination of factors, future sea ice lows are likely to also be triggered by several co-occurring events. Changes in the frequency and intensity of such events is not clear. Arctic cyclones are not expected to become more frequent in the future (Crawford & Serreze, 2017), but both Arctic and Antarctic cyclones have intensified (Zhang et al., 2023; Chemke et al., 2022). Heatwaves, especially marine heatwaves, have increased in frequency and intensity in the Arctic (Huang et al. 2021) and could therefore contribute to more sea ice lows."
* * *
**Comment 11**

L120: "lateral melting through ice floe size parametrisation": suggest citing Lüpkes et al. (2012) for this.
* * *
**Response:**

We will add the reference.
* * *
**Comment 12**

L121: "level-ice melt ponds": suggest adding a citation (Hunke et al., 2013) for this too as it may not be obvious to all readers that "level-ice" refers to a specific parameterisation choice.
* * *
**Response:**

We will add the reference.
* * *
**Comment 13**

L140: I am satisfied with the justification that the bias in the forcing, and hence simulation, can be overlooked by removing the mean/trend and examining anomalies. However, I wonder why the authors decided to use ERA5 as atmospheric forcing despite known issues (the authors themselves cite a study from 2019) with temperature biases in the polar regions specifically. What is the benefit of using ERA5 over another reanalysis, for example? It would be good if the authors could add a bit more here to justify the choice of ERA5, even if it is just on practical grounds such as data availability or suitable resolution.
* * *
**Response:**

We use ERA5 mainly for practical reasons, as it covers the right period at a proper horizontal and temporal resolution; it is also a well-validated, often used reanalysis product for forcing ocean-sea ice models. Finally, it is updated in near-real time, which allows for operational simulation of recent events, for other projects. It is worth noting that other reanalysis products exhibit similar biases in polar regions, for the same reasons, and that ERA5 seems to behave rather better than most other products (Batrak & Müller 2019).

We propose to clarify this by adding, l.122: "[...] extracted from the ECMWF ReAnalysis v5 (ERA5, Hersbach et al., 2020), a well validated, regularly updated forcing set at the 1/4° horizontal resolution "
and by modifying, l. 145: "[...] due to lack of representation of snow over ice in the models used to produce most reanalysis products, including ERA5 (Batrak and Müller, 2019)."
* * *
**Comment 14**

L164–167: I think this important justification for using a model analysis would make more sense at the beginning of section 2.1 or perhaps somewhere towards the end of section 1, because at this stage the model has already been introduced/described.
* * *
**Response:**
We agree, we will move those 4 sentences at the beginning of the last paragraph of section 1; it will actually provide a great transition with the previous paragraph.
* * *
**Comment 15**

L183: do you mean "become clearer in Section 3" (this seems to be discussed there, on L232–233, L282–283).
* * *
**Response:**
Yes, this was a typo, thank you.
* * *
**Comment 16**

Figure 2: I'm not sure this figure adds much and is potentially a little confusing. The arrows point upwards for positive terms which corresponds to increasing ice mass. This works visually for, e.g., basal growth (mass moves from the ocean to ice), but it does not work for the snow–ice term (the arrow is consistently drawn upward for the sign convention, but visually this is like mass moving from ice to snow). The transport term makes sense visually but does not match the upward/downward = positive/negative flux. I realise this is explained in the caption, but overall the diagram itself does not consistently illustrate the sign convention or the physical processes (the latter being the more intuitive use of such a figure). I would suggest removing as the budget decomposition is fairly standard and in any case it is intuitive enough which terms increase or decrease sea ice mass. As for the sign convention, this is stated clearly enough in the main text and helpfully repeated in the captions of Figs. 4–5, so again I do not think the figure is needed.
* * *
**Response:**
We understand the reviewer's point and agree that arrows referring to physical processes is also often used for such figures and can seem more intuitive. However, as this figure provides a visual summary of all the processes we investigate in this paper, and not all readers will be familiar with all the sea-ice mass balance terms, we think that it is useful to keep it. Moreover, comments from the other reviewers make us think that the sign convention is not so obvious, and that this schematic illustration can help clarify it. Could the editor provide a third opinion on whether we should keep this figure or not?
* * *
> **Comment 17**
>
> Figure 3: is it possible to arrange this horizontally, as in Fig. 4?

**Response:**

A new horizontal version will replace the previous one. See Figure R4.
* * *
> **Comment 18**
>
> L225: "slightly higher": some estimation/quantification should be given (e.g., is it a few percent?)

**Response:**

On top of a different time period, Keen et al. 2021 restrain their study domain to the high Arctic (similar but not identical to the sum of our Barents-Kara Sea, Siberian seas, Central Arctic, Beaufort Sea and Chukchi sea north of Bering Strait sectors). Therefore, we cannot give a direct comparison. Nonetheless, we propose to provide an estimate of the difference and will rephrase, l. 225: "The absolute magnitudes are 25 to 40 % higher than those of the ensemble mean reported in Keen et al. (2021) for a similar though not identical domain. This could be due to differences in the study domains and periods of interest, and to the overestimated amplitude of the seasonal cycle in ice extent in the model used here."
* * *
> **Comment 19**
>
> L226–227: it might also be related to the absence of atmospheric feedbacks? Anyway, this is a relevant limitation of the methodology which should be mentioned in the discussion with some consideration of potential impacts on the results.

**Response:**

This is a good point that we will incorporate by adding a sentence, l. 227: "It could also be related to the lack of atmospheric feedbacks in our model, as coupling the ocean and sea ice to the atmosphere (as is the case in the CMIP6 models) should dampen the sea ice response to atmospheric or oceanic perturbations." Regarding those limitations in general, they are already mentioned in the Discussion (l. 507-530), when we discuss how the biased mean sea ice state could impact the amplitude of the mass fluxes.
* * *
> **Comment 20**
>
> L246–247: could you expand on/clarify this comparison? The study cited is examining the sensitivity of a similar model's mass balance to the discretisation of the ice thickness distribution. The statement that "some quantitative discrepancies exist" is vague.

**Response:**

Here again, we will expand this comparison, as suggested by the reviewer. please note that those comparisons have limited values, as the backbone model is the same so the data are not independent, and the methodologies differ on the area and period of interest. Moreover, our sea ice volume climatological seasonal cycle seems closer to the GIOMAS reanalysis than that of Massonnet et al. (2019). We will rephrase this comparison as follow, l. 247: "They are also in good qualitative agreement with the S1.05 experiment from Massonnet et al. (2019), though some quantitative

discrepancies exist. For example, our model exhibits a lower basal melt and frazil ice growth in the summer and fall seasons, likely due to a different domain definition and sea ice volume state in our simulation. Winter and spring values agree very well with the Massonnet et al. (2019) study."
* * *
**Comment 21**

L248–251: This paragraph comparing the Arctic and Antarctic is quite short. If there is not much else to say, I would suggest merging it with the end of the previous (Antarctic) paragraph. Alternatively, it could be expanded by drawing some conclusions, such as expecting to see more sea ice lows in certain regions of the Arctic but more uniformly around Antarctica. Also, why is it interesting to note similarity of the Greenland Sea sector with the Antarctic? Presumably, it means we might expect similar mechanisms for sea ice lows occurring in those sectors.

**Response:**

This paragraph is indeed short and we will merge it with the previous one. Anticipating the results of section 4 based on the climatology might confuse the reader and we therefore opt to not expand this paragraph. But we agree with the reviewer that this climatology comparison hints toward more heterogeneity in the Arctic sea ice low spatial distribution and believe this is more or less what we see when comparing 2007 and 2012. We also agree on the fact that Greenland Sea exhibits mechanisms similar to the Antarctic. We could expect that with Arctic Atlantification, more similarities might be observed in a near future in other parts of Arctic. But we refrain to mention this in the manuscript, as this might seem too speculative.
* * *
**Comment 22**

L311–313: Figure S1 is plotting May, not March.

**Response:**

Thank you, this was a typo.
* * *
**Comment 23**

L334: "consistent with observations": I think a reference is warranted here to justify this statement (as is done later for the Antarctic case, L410).

**Response:**

Surprisingly enough, no peer-reviewed map of the sea ice state in September 2012 could be found. We will include a footnote referring to the following webpage: https://nsidc.org/sea-ice-today/analyses/arctic-sea-ice-extent-settles-record-seasonal-minimum
* * *
**Comment 24**

L350: "Yet, no significant influence of the cyclone is visible in the model outputs (not shown)": this is vague and unsubstantiated, and also slightly confusing given that the next few lines discuss indirect effects on the ocean heat content over the Beaufort sea. Perhaps, in the quoted line, the authors meant "direct", rather than "significant", impacts? Some comment on what impacts on the model outputs were checked but not seen should be added here.

**Response:**
When mentioning "significant", we meant an influence that could be first order in determining the September sea ice mass. We have looked at time series of mass flux terms (including surface and basal melt) and while some increased basal melt over the Beaufort and Chukchi Seas and ice import in Central Arctic sectors can be seen in early August (coinciding with the cyclone), the magnitude of those anomalies are within the range of the variability of those terms over the whole season, and no traces of those anomalies can be seen on the equivalent time series for the whole Arctic. We will clarify this in the text by rephrasing, l. 350: "In the model, basal melt in Beaufort Sea and Chukchi Sea and ice import in the Central Arctic both show a small increase in early August, coinciding with the cyclone, but do not result in a visible anomaly in the respective terms integrated over the whole Arctic."
* * *
**Comment 25**

L375: "match well with satellite observations of sea ice concentration anomalies": but the plot is of sea ice thickness? Similar for L409–410.
* * *
**Response:**
Indeed, we skipped a step by comparing the sea ice thickness anomalies with the observed sea ice concentration. While we do expect both to show similar spatial patterns, we did not clarify this in the initial manuscript. We will correct this omission by adding maps of sea ice concentration anomalies with observed sea ice extent contours in the SIs (Figure R2). We will adapt the main text to refer to this figure, as well as to the background sea ice thickness anomalies maps.
* * *
**Comment 26**

L401: in section 4.3, Fig. 5.b is not cross referenced anywhere (in fact, Fig. 5.b is not referred to anyway in the manuscript).
* * *
**Response:**
Thank you, this was a mistake. We will add a reference to it at the end of l. 408.
* * *
**Comment 27**

L425–426: The phrase "more difficult to analyse and understand" should be removed as it is vague (no reason is given) and the next few sentences seem to provide analysis of this sector anyway. Also, I'm not convinced that the anomalies there are "small compared to the other sectors", assuming the units of each inset panel of Fig. 5.b are the same: for example, the total mass flux (black bar) is about 130 Gt, certainly larger in magnitude than that of the Ross–Amundsen sector (less than 100 Gt), and the snow-to-ice formation (pale blue bar) is larger in magnitude than any of the other sectors.
* * *
**Response:**
This is a good point, we will remove this sentence and modify the one after by (l. 426): "In the East Antarctic sector, a small positive mass flux anomaly [...]".

[Figure]

Figure R2: Suggested figure to include in supplementary information, showing the modelled sea ice concentration anomalies (colours) and sea ice extent (grey contour), to compare with observation-based sea ice extent from NSIDC CDR product (black contour), for all four sea ice lows detailed in the manuscript.
* * *
**Comment 28**

L544: "direct": did you mean "indirect"? (I would have thought the "direct" effect of the atmosphere would be via surface melt/growth, and the "indirect" effect via eventual influence on basal melt/growth as described in the text).
* * *
**Response:**

We consider a change in the heat conductive flux as a direct effect of the atmosphere forcing, in opposition with an increased basal melt due to warmer waters that would have been heated by the atmosphere in leads or in the marginal ice zone, for example. We agree this is slightly subjective, but the basal melt would respond nearly immediately to a change of the temperature profile (and therefore the conductive flux) induced by atmospheric conditions and would not be subject to another medium transferring the heat (unless we consider the ice as an intermediate medium, which could also make sense). We propose to keep the "direct" wording, as the context of this paragraph is to describe the role of the atmosphere, as opposed to the role of the ocean, in the basal heat budget.
* * *
**Comment 29**

L551: important to note that the studies cited here are about the longer timescale (multi-decadal) impact of ocean heat transport on sea ice, not about yearly lows.
* * *
**Response:**

Good point, we will add this note, l.551: "[...] a number of recent studies have shown

the importance of oceanic heat transport to explain ice melt over multi-decadal time scales"
* * *
**Comment 30**

Figures 4 and 5: these are very nice figures, but a couple of small suggestions: a different colour map for the sea ice thickness might be helpful visually, as it currently clashes with the colour choices for the budget terms. Also, in the captions, I suggest noting that the units of each inset are the same (assuming they are?), e.g., "the scales of the y-axes differ among panels, but the units are always Gt)".
* * *
**Response:**

Thank you. We will change the colormap of the thickness anomalies (brown to green), hoping this clarifies the distinction (see new version in Figs R5 & R6). We will also mention in the caption that units and y-axis labels are the same for all inset panels.

**References**

Bitz, C. M. and Roe, G. H.: A Mechanism for the High Rate of Sea Ice Thinning in the Arctic Ocean, Journal of Climate, 17, 3623–3632, `https://doi.org/10.1175/1520-0442(2004)017<3623:AMFTHR>2.0.CO;2`, 2004.

Chemke, R., Ming, Y., and Yuval, J.: The intensification of winter mid-latitude storm tracks in the Southern Hemisphere, Nat. Clim. Chang., 12, 553–557, `https://doi.org/10.1038/s41558-022-01368-8`, 2022.

Docquier, D., Massonnet, F., Barthélemy, A., Tandon, N. F., Lecomte, O., and Fichefet, T.: Relationships between Arctic sea ice drift and strength modelled by NEMO-LIM3.6, The Cryosphere, 11, 2829–2846, `https://doi.org/10.5194/tc-11-2829-2017`, 2017.

Olason, E. and Notz, D.: Drivers of variability in Arctic sea-ice drift speed, Journal of Geophysical Research: Oceans, 119, 5755–5775, `https://doi.org/10.1002/2014JC009897`, 2014.

Smith, K. E., Sen Gupta, A., Amaya, D., Benthuysen, J. A., Burrows, M. T., Capotondi, A., Filbee-Dexter, K., Frölicher, T. L., Hobday, A. J., Holbrook, N. J., Malan, N., Moore, P. J., Oliver, E. C. J., Richaud, B., Salcedo-Castro, J., Smale, D. A., Thomsen, M., and Wernberg, T.: Baseline matters: Challenges and implications of different marine heatwave baselines, Progress in Oceanography, 231, 103404, `https://doi.org/10.1016/j.pocean.2024.103404`, 2025.

Zhang, X., Tang, H., Zhang, J., Walsh, J. E., Roesler, E. L., Hillman, B., Ballinger, T. J., and Weijer, W.: Arctic cyclones have become more intense and longer-lived over the past seven decades, Commun Earth Environ, 4, 348, `https://doi.org/10.1038/s43247-023-01003-0`, 2023.

**Figures: new versions**

[Figure]

Figure R3: Figure 1, new version. Modified caption:
Comparison of observations-based products (black) versus model sea ice extent (red) and volume (blue). Observations-based products are the satellite-based NOAA/NSIDC Climate Data Record (CDR) for sea ice extent and the PIOMAS and GIOMAS reanalyses products for sea ice volume. Mean seasonal cycle of sea ice extent for (a) Arctic and (d) Antarctic. Shading indicates one standard deviation. Minimum sea ice extent anomalies relative to the mean seasonal cycle for (b) September in Arctic and (e) February in Antarctic. Minimum sea ice volume anomalies relative to the mean seasonal cycle for (c) September in Arctic and (f) February in Antarctic; correlations between observations and model are given in the top-right corner of each panel for the minima comparisons (panels b, c, e and f). Note that the time values include the month, meaning that the point for e.g. September 2000 is closer to the x-tick value 2001 than 2000. For more robust comparisons, all mean seasonal cycles used in this figure are calculated over the whole available satellite period (1979-2023).

[Figure]

Figure R4: Figure 3, new version.

[Figure]

Figure R5: Figure 4, new version.

[Figure]

Figure R6: Figure 5, new version.